# Discovery of hierarchical representations for efficient planning

**Momchil S. Tomov**[1,2]*, **Samyukta Yagati**[3], **Agni Kumar**[3], **Wanqian Yang**[4], **Samuel J. Gershman**[2]

**1** Program in Neuroscience, Harvard Medical School, Boston, Massachusetts, United States of America, **2** Department of Psychology and Center for Brain Science, Harvard University, Cambridge, Massachusetts, United States of America, **3** Department of Electrical Engineering and Computer Science, Massachusetts Institute of Technology, Cambridge, Massachusetts, United States of America, **4** School of Engineering and Applied Sciences, Harvard University, Cambridge, Massachusetts, United States of America

\* mtomov@g.harvard.edu

**Data Availability Statement:** All data and code files for all simulations and experiments are freely available in the Github repository at this URL: https://github.com/tomov/chunking.

## Abstract

We propose that humans spontaneously organize environments into clusters of states that support hierarchical planning, enabling them to tackle challenging problems by breaking them down into sub-problems at various levels of abstraction. People constantly rely on such hierarchical presentations to accomplish tasks big and small—from planning one's day, to organizing a wedding, to getting a PhD—often succeeding on the very first attempt. We formalize a Bayesian model of hierarchy discovery that explains how humans discover such useful abstractions. Building on principles developed in structure learning and robotics, the model predicts that hierarchy discovery should be sensitive to the topological structure, reward distribution, and distribution of tasks in the environment. In five simulations, we show that the model accounts for previously reported effects of environment structure on planning behavior, such as detection of bottleneck states and transitions. We then test the novel predictions of the model in eight behavioral experiments, demonstrating how the distribution of tasks and rewards can influence planning behavior via the discovered hierarchy, sometimes facilitating and sometimes hindering performance. We find evidence that the hierarchy discovery process unfolds incrementally across trials. Finally, we propose how hierarchy discovery and hierarchical planning might be implemented in the brain. Together, these findings present an important advance in our understanding of how the brain might use Bayesian inference to discover and exploit the hidden hierarchical structure of the environment.

## Author summary

Human planning is hierarchical. Whether planning something simple like cooking dinner or something complex like a trip abroad, we usually begin with a rough mental sketch of the goals we want to achieve ("go to Spain, then go back home"). This sketch is then progressively refined into a detailed sequence of sub-goals ("book flight", "pack luggage"), sub-sub-goals, and so on, down to the actual sequence of bodily movements that is much

**Funding:** Momchil Tomov and Samuel Gershman were funded by the Toyota corporation (https://www.toyota.com/), the Office of Naval Research (Award N000141712984; https://www.onr.navy.mil/), the Multi-University Research Initiative Grant (Office of Naval Research Grant N00014-17-1-2961; https://www.onr.navy.mil/), and the National Institutes of Health (CRCNS 1R01MH109177; https://www.nih.gov/). The funders had no role in study design, data collection and analysis, decision to publish, or preparation of the manuscript.

**Competing interests:** The authors have declared that no competing interests exist.

more complicated than the original plan. Efficient planning therefore requires knowledge of the abstract high-level concepts that constitute the essence of hierarchical plans. Yet how humans learn such abstractions remains a mystery. In this study, we show that humans spontaneously form such high-level concepts in a way that allows them to plan efficiently given the tasks, rewards, and structure of their environment. We also show that this behavior is consistent with a formal model of the underlying computations, thus grounding these findings in established computational principles and relating them to previous studies of hierarchical planning. We believe our results pave the way for future studies to investigate the neural algorithms that support this essential cognitive ability.

## Introduction

Imagine you have a sudden irresistible craving for your favorite ice cream that is only made by a boutique ice cream shop in Lugo, Spain. You must get there as soon as physically possible. What would you do? When faced with this unusual puzzle, most people's first response is that they will look up a flight to Spain. When asked what they would do next, most people say that they would order a taxi to the airport, and when questioned further, that they would walk to the taxi pickup location. Importantly, nobody says or even thinks anything like "I will get up, turn left, walk five steps, etc.", or even worse, "I will contract my left quadricep, then my right one, etc.". This example illustrates hierarchical planning (Fig 1): people intuitively reason at the appropriate level of abstraction, first sketching out a plan in terms of transitions between high-level *states* (in this case, countries), which is subsequently refined in progressively lower

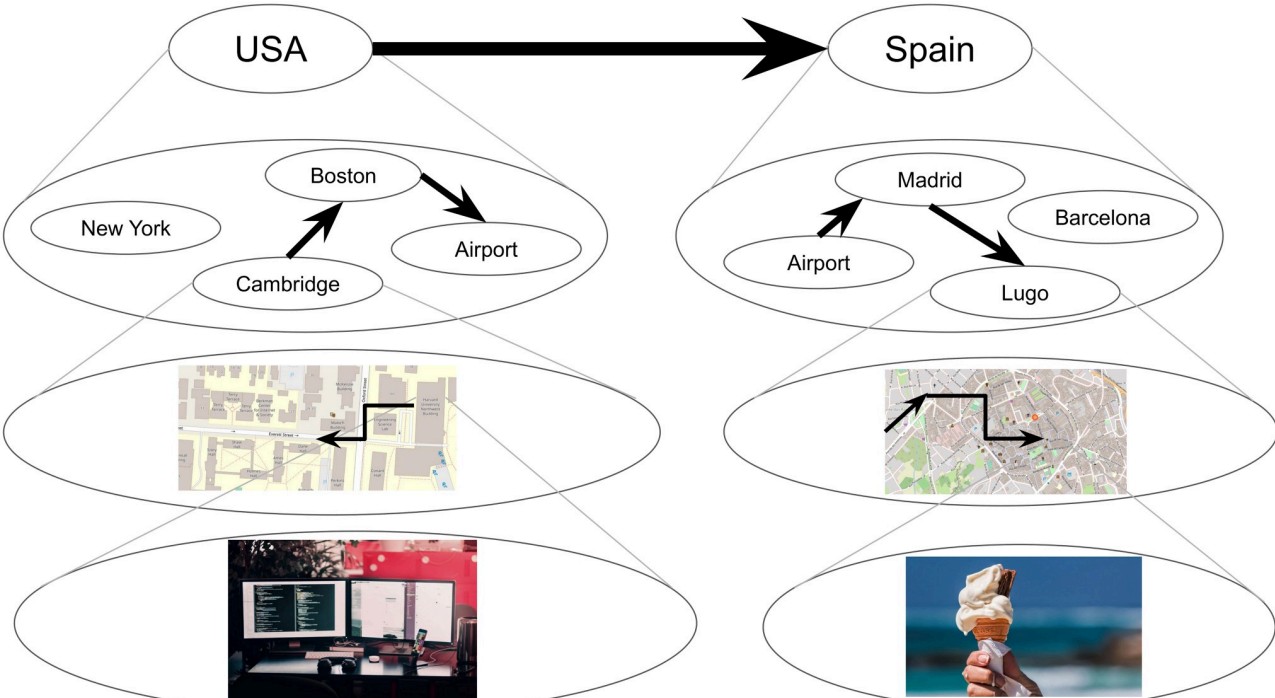

**Fig 1. Example of hierarchical planning.** How someone might plan to get from their office in Cambridge to their favorite ice cream shop in Lugo, Spain. Circles represent states and arrows represent actions that transition between states. Each state represents a cluster of states in the lower level. Thicker arrows indicate transitions between higher-level states, which often come to mind first.

levels and specific steps [1]. This is often done in an online fashion, with details being resolved on-the-fly as the plan is being executed (for example, you would not ponder what snack to buy at the airport before actually getting there). This ability of humans [2] and animals [3] to organize their behavior hierarchically allows them to flexibly pursue distant goals in complex environments, even for novel tasks that they may have never encountered previously.

The study of hierarchical behaviors has deep roots in psychology and neuroscience [4]. Much work has been done to characterize the emergence of such behaviors in humans and animals, often focusing on the acquisition of temporally extended action sequences or *action chunks* that unfold over different time scales. Action chunking occurs after extensive training, involves a specific set of brain regions, and is thought to be essential for pursuing long-term goals and planning [5, 6].

Yet in order to leverage action chunks for planning, an agent must also be equipped with a hierarchical representation of the environment, with clusters of states or *state chunks* representing parts of the world at different levels of abstraction. In our ice cream example, the current country, city, neighborhood, building, room, and location within the room are all valid representations of the agent's current state and are all necessary in order to plan effectively. Correspondingly, research shows that people spontaneously discover hierarchical structure [1, 7], that the discovered hierarchies are consistent with formal definitions of optimality [8], and that people use these hierarchies to plan [2]. Some of these studies have uncovered distinct neural correlates of *high-level states*, which are sometimes referred to as state clusters, communities, contexts, abstract states, or state chunks [2, 7]. Research has also begun to uncover the neural correlates of hierarchical planning and action selection [2, 9]. Yet despite these advances, the computational mechanisms underlying the discovery of such hierarchical representations remain poorly understood.

In this study, we propose a Bayesian model of hierarchy discovery for planning. Drawing on the structure learning literature [10] and on concepts developed in robotics [11], the model provides a normative account of how agents with limited cognitive resources should chunk their environment into clusters of states that support efficient planning. The central novel contributions of this paper are both empirical and theoretical. Our main empirical contribution is to show that the distribution of tasks (experiments one through five) and rewards (experiments six and seven) in the environment can influence the inferred state chunks, whereas past studies have focused exclusively on the effects of environment topology. Our main theoretical contribution is to unify these and previous findings under a single normative model that explains why these phenomena occur, and that encompasses a class of process models that could be leveraged to investigate the implementational details of state chunking in the brain.

In simulations one through five, we demonstrate that the model accounts for previously reported behavioral effects of the environment topology, such as detection of transitions across state clusters [7], identification of topological bottlenecks [8], preference for routes with fewer state clusters [8], and slower reaction times to transitions that violate topological structure [12]. Additionally, the model makes specific predictions about the way in which the distributions of tasks and rewards constrain hierarchy discovery, which in turn constrains planning. We test these novel predictions in a series of eight behavioral experiments. Experiment one shows that the distribution of tasks encountered in the environment can induce different state clusters, even when the topological structure of the environment does not promote any particular clustering. Experiment two shows how this in turn could lead to either improved or hampered performance on novel tasks. Experiment three examines the progression of inferred hierarchies as a function of which tasks participants have seen so far and reveals that participants are sensitive to changes in both the uncertainty and the mode of the posterior distribution over hierarchies. Experiments four and five replicate the results of experiments one and

two in a fully visible environment, showing that the effects cannot be explained by incorrect inferences about topological structure. Experiment six shows how rewards generalize within state clusters, while experiment seven shows how rewards can induce clusters that constrain planning even in the absence of state clusters. Finally, experiment eight (see S1 Appendix) suggests that people explore in a way that maximally reduces uncertainty about the hierarchy, implying that people consider a probability distribution over hierarchies rather than a single hierarchy. Together, these results provide strong support for a Bayesian account of hierarchy discovery, according to which the brain implicitly assumes that the environment has a hidden hierarchical structure, which is rationally inferred from observations. No existing theory of hierarchy discovery can account for all of these effects (Table 2).

## Results

### Theoretical background

The question of how agents should make rewarding decisions is the subject of reinforcement learning (RL; [13]). With a long history crisscrossing the fields of psychology, neuroscience, and artificial intelligence, RL has made major contributions to explaining many human and animal behaviors [14], the neural circuits underlying these behaviors [15], and allowing artificial agents to achieve human-level performance on tasks that were previously beyond the capabilities of computers [16]. Approaches rooted in RL therefore offer promising avenues to understanding decision making and planning.

Most RL algorithms assume that that an agent represents its environment as a Markov decision process (MDP), which consists of states, actions and rewards that obey a particular conditional independence structure. At any point in time, the agent occupies a given state, which could denote, for example, a physical location such as a place in room, a physiological state such as being hungry, and abstract state such as whether a subgoal has been achieved, or a complex state such as a conjunction of such states. The agent can transition between states by taking actions such as moving or eating. Some states deliver rewards, and it is assumed that the agent aims to maximize reward. In an MDP, the transitions and rewards are assumed to depend only on the current state and action.

Following previous authors [2, 8], we assume for simplicity that states are discrete and that transitions between states are bidirectional and deterministic (although these restrictions could be relaxed; see Discussion). In this case, states and actions could be represented by an undirected graph $G$ in which the vertices correspond to states and the edges correspond to actions. We will use the graph $G$ in place of the transition function $T$ that is traditionally used to characterize MDPs.

In graph theoretic notation, $G = (V, E)$, where

- $V$ is the set of vertices (or nodes) such that each node $v \in V$ corresponds to a state that the agent can occupy, and

- $E$: {$V \times V$} is a set of edges such that each edge $(u, v) \in E$ corresponds to an action that the agent can take to transition from state $u$ to state $v$ or vice versa.

In the following analysis, we use the terms *node* and *state* interchangeably. We also treat edges, actions, and transitions as equivalent. For simplicity, we restrict our analysis to unweighted graphs, which is equivalent to assuming that all actions carry the same cost and/or take the same amount of time (our analysis extends straightforwardly to weighted graphs; see Discussion). We also assume the agent has learned $G$, which is equivalent to model-based RL in which the agent learns the transition function.

The task of planning to get from a starting state $s$ to a goal state $g$ efficiently can thus be framed as finding the shortest path between nodes $s$ and $g$ in $G$. This is a well-studied problem in graph theory and the optimal solution in this setup is the breadth first search (BFS) algorithm [17]. BFS works by first exploring the neighbors of $s$, then exploring the neighbors of those neighbors, and so on until reaching the goal state $g$. States whose neighbors haven't been explored yet are maintained in an active queue, with states getting removed from one end of the queue as soon as their neighbors are added to the other end. Intuitively, this corresponds to a forward sweep that begins at $s$ and spreads in all directions across the edges until reaching $g$, akin to the way in which water might spread in a network of pipes. Its simplicity and performance guarantees have made BFS a standard tool for planning in classical artificial intelligence [18].

However, the time and memory that BFS requires is proportional to the number of states (assuming an action space of constant size) since the size of the active queue and the potential length of the plan grow linearly with the size of the state space. Formally, the time and memory complexity of BFS is $O(N)$, where $N = |V|$ is the total number of states. In environments in which real-world agents operate, this number can be huge; in the realm of navigation alone, there could be billions of locations where a person has been or may want to go. Assuming that online computations such as planning involve systems for short-term storage and symbol manipulation, this far exceeds the working memory capacity of people who can only accommodate a few items (note that we assume the graph is already stored in a different system for long-term storage of relational information, such as the hippocampus) [19]. Furthermore, even without such working memory limitations, artificial agents such as robots would still take a long time to plan the route before they can take the first step. When using a naive or "flat" representation in which the agent plans over low-level actions (for example, individual steps or even joint actuator commands), computing a plan is at least as complicated as actually executing it (Fig 2A), and in reality the complexity could be much larger.

To overcome this limitation, work in the field of robotics has led to the development of data structures and algorithms for hierarchical planning [11]. Similar ideas have been put forward in other fields; see Discussion. The key idea is that an agent can group neighboring states from the flat low-level graph $G$ into state clusters (state chunks), with each cluster represented by a single node in another graph $H$ (the high-level graph), which will be smaller and hence easier to plan in. When tasked to get from state $s$ to state $g$ in $G$, the agent can first plan in the high-level graph $H$ and then translate this high-level plan into a low-level plan in $G$. Crucially, after finding the high-level path in $H$, the agent only needs to plan in the current state cluster in $G$, that is, it only needs to plan how to get to the next cluster (Fig 2B), and then repeat the process in the next cluster, and so on, until reaching the goal state in the final cluster.

This can drastically reduce the working memory requirements of planning, since the agent only needs to keep track of the (much shorter) path in $H$ and the path in the current state cluster in $G$. Importantly, this also reduces planning time, allowing the agent to begin making progress towards the goal without computing the full path in $G$—the agent can now follow the high-level plan in $H$ and gradually refine it in $G$ on-the-fly, during execution. This approach can be applied recursively to deeper hierarchies in which high-level states are clustered in turn into even higher-level states, and so on until reaching a single node at the top of the hierarchy that represents the entire environment (Fig 2C). Planning using such a hierarchical representation can be orders of magnitude more efficient than "flat" planning, and also accords with our intuitions about how people plan in everyday life.

A particular instantiation of this form of hierarchical planning is the hierarchical breadth first search (HBFS) algorithm, which is a natural extension of BFS (see Methods). It can be shown that with an efficient hierarchy of depth $L$ (that is, consisting of $L$ graphs, with each

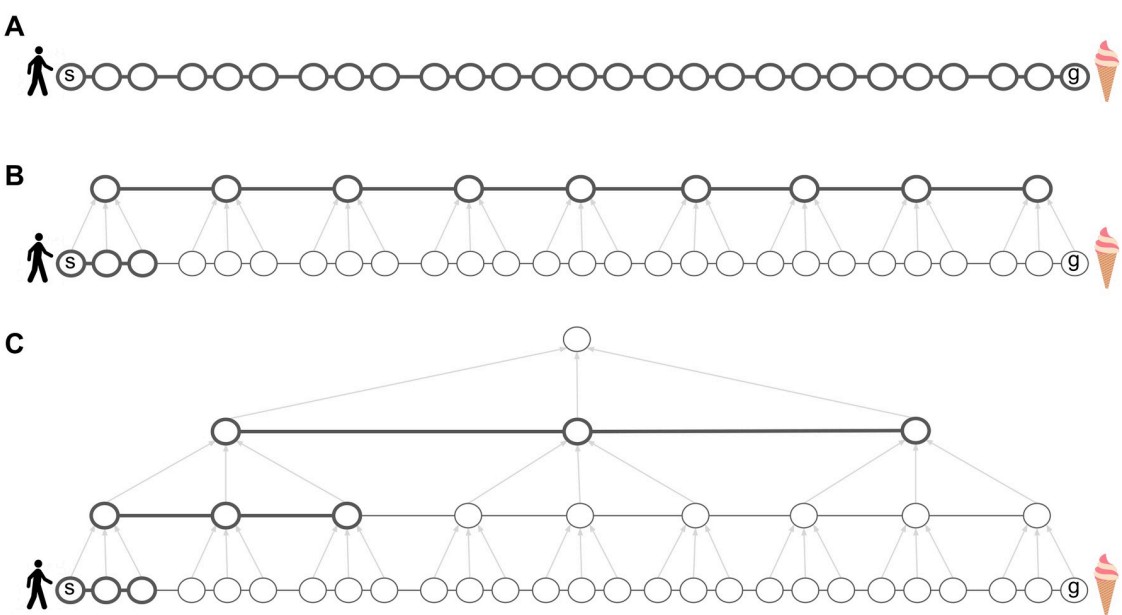

**Fig 2. Hierarchical representations reduce the computational costs of planning.** A. Planning in the low-level graph *G* takes at least as many steps as actually executing the plan. All nodes and edges are thick, indicating that they must all be considered and maintained in short-term memory in order to compute the plan. B. Introducing a high-level graph *H* alleviates this problem. At any given time during plan execution, the agent only needs to consider the high-level path and the low-level path leading to the next cluster, recomputing the latter on-the-fly. Gray arrows indicate cluster membership. C. The hierarchy can be extended recursively, further reducing the time and memory requirements of planning.

graph representing clusters in the lower graph), the time and memory complexity of HBFS is $O(\sqrt[L]{N}+L)$ [11]. Thus in a graph with $N = 1,000,000,000$ states and a hierarchy $L = 9$ levels deep, HBFS will only require on the order of 19 memory and time units to compute a plan, compared to 1,000,000,000 time and memory units for BFS, or any other flat planner. Note that while executing the plan would still take $O(N)$ time, HBFS quickly computes the first few actions in the right direction, and can then be applied repeatedly to keep computing the following actions in an online fashion. While it may seem that this hierarchical scheme simply transfers the burden to the long-term storage system, which now needs to remember *L* graphs instead of one, it can be shown that the storage requirements of an efficient hierarchical representation are $O(N)$ [11], comparable to those of a flat representation of *G* alone. Following previous authors, [8, 20], we restrict our analysis to $L = 2$ levels for simplicity, noting that our approach extends straightforwardly to deeper hierarchies (see Discussion).

However, in order to reap the benefits of efficient planning, the hierarchical representation necessarily imposes a form of lossy compression. In particular, each successive graph in the hierarchy loses some of the detail present in the lower level graph. This could lead to hierarchical plans that correspond to suboptimal paths in *G*. For example, in Fig 3A, there is a direct edge that can take the agent from starting state *s* to go state *g* in a single action. However, since the edge is not represented by the high-level graph *H*, HBFS will compute a detour through the state cluster *w*, akin to real-life situations in which people prefer going through a central location rather than taking an unfamiliar shortcut.

Since finding the shortest path will not always be possible, some hierarchies will tend to yield better paths than others. The challenge for the agent is then to learn an efficient hierarchical representation of the environment that facilitates fast planning across many tasks, without placing an overwhelming burden on working memory and long-term memory systems. How

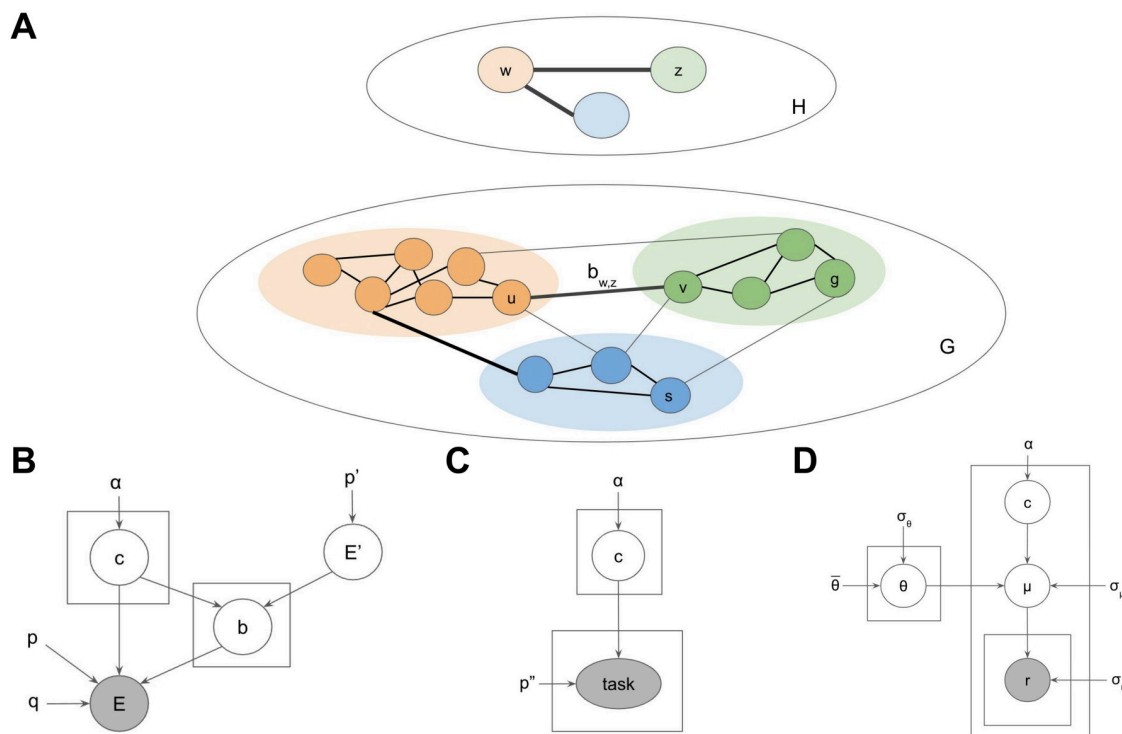

**Fig 3. Generative model for environments with hierarchical structure.** A. Example low-level graph $G$ and high-level graph $H$. Colors denote cluster assignments. Black edges are considered during planning. Gray edges are ignored by the planner. Thick edges correspond to transitions across clusters. The transition between clusters $w$ and $z$ is accomplished via the bridge $b_{w,z} = (u, v)$. B. Generative model defining a probability distribution over hierarchies $H$ and environments $G$. Circles denote random variables. Rectangles denote repeated draws of a random variable. Arrows denote conditional dependence. Gray variables are directly observed by the agent. Uncircled variables are constant. $c$, cluster assignments; $p'$, graph density of $H$; $E'$, edges in $H$; $E$, edges in $G$; $b$, bridges connecting the clusters; $p$, within-cluster graph density in $G$; $q$, cross-cluster graph density penalty in $G$. Refer to main text for variable definitions. C. Incorporating tasks into the generative model. The rest of the generative model is omitted for clarity. $p''$, cross-cluster task penalty; task $= (s, g)$, task as pair of start-goal states. D. Incorporating rewards into the generative model. The rest of the generative model is omitted for clarity. $\bar{\theta}$, average reward for $G$; $\sigma_\theta$, standard deviation of that average; $\theta$, average cluster rewards; $\sigma_\mu$, standard deviation around that average; $\mu$, average state rewards; $\sigma_r$, standard deviation around that average; $r$, instantaneous state rewards.

agents accomplish this in the real world is the central question of this paper, to which we propose a solution in the following section.

## A Bayesian model of hierarchy discovery

Our proposal assumes two main computational components:

1. An online planner that can flexibly generate plans and select actions on-demand with minimal time and memory requirements.

2. An off-line (and possibly computationally intensive) hierarchy discovery process that, through experience, incrementally builds a representation of the environment that the planner can use.

The focus of this paper is the second component, which must satisfy the constraints imposed by the first component. For the first component, we use HBFS in order to link the hierarchy to behavior, noting that any generic hierarchical planner will make similar predictions. Note that we assume only the online planner is constrained by working memory

limitations and time demands—as in the ice cream example, the high-level sketch of a plan is often computed within seconds of the query. In contrast, the hierarchical representation that supports this computation—a rich abstraction of the world, with knowledge of particular locations that belong to cities that belong to countries connected by flights—has been refined through years of experience and is deeply ingrained in long-term memory.

One approach to deriving an optimal hierarchy discovery algorithm would be to define the constraints of the agent, such as memory limitations and computational capacity, its utility function, and the constraints of the environment, such as the expected structure and tasks. Here we adopt an alternative approach, motivated by the literature on structure learning which has been used to successfully account for a wide range of phenomena in the animal learning literature [10]. The key idea is that the environment is assumed to have a hidden hierarchical structure that is not directly observable, which in turn constrains the observations the agent can experience. The agent can then infer this hidden hierarchical structure based on its observations and use it to plan efficiently. Assuming that some hierarchies are *a priori* more likely than others, this corresponds to a generative model for environments with hierarchical structure, which the agent can invert to uncover the underlying hierarchy based on its experiences in the environment.

Formally, we represent the observable environment as an unweighted, undirected graph $G = (V, E)$ and the hidden hierarchy as $H = (V', E', c, b, p', p, q)$, where:

- $V$ is the set of low-level nodes or states, corresponding to directly observable states in the environment,

- $E$: $\{V \times V\}$ is the set of edges, corresponding to possible transitions between states via taking actions,

- $V'$ is the set of high-level nodes or states, corresponding to clusters of low-level states,

- $E'$: $\{V' \times V'\}$ is the set of high-level edges, corresponding to transitions between high-level states,

- $c$: $V \rightarrow V'$ are the *cluster assignments* linking low-level states to high-level states,

- $b$: $E' \rightarrow E$ are the *bridges* which link high-level transitions back to low-level transitions,

- $p' \in [0, 1]$ is the density of the high-level graph,

- $p \in [0, 1]$ is the within-cluster density of $G$,

- $q \in [0, 1]$ is the across-cluster density penalty of $G$.

Here we use the term *graph density* to denote the probability that a pair of nodes are connected by an edge. Together, $(V', E')$ define the high-level graph, which we also refer to as $H$ for ease of notation. Each low-level state $u$ is assigned to a cluster $w = c_u$. Each high-level edge $(w, z)$ has a corresponding low-level edge (the bridge) $(u, v) = b_{w,z}$, such that $c_u = w$ and $c_v = z$ (see Fig 3A). Bridges (sometimes referred to as bottlenecks or boundaries) thus specify how different clusters are connected in $H$. Bridges are the only cross-cluster edges considered by the hierarchical planner; all other edges between nodes in different clusters are ignored, leading to lossy compression that improves planning complexity but could lead to suboptimal paths. In contrast, all edges within clusters are preserved. The purpose of the cluster assignments $c$ is to translate the low-level planning problem in $G$ into an easier high-level planning problem in $H$. The purpose of the bridges is to translate the solution found in $H$ back into a low-level path in $G$. For a detailed description of how HBFS uses the hierarchy to plan, see the Methods section. For simplicity, we only allow a single bridge for each pair of clusters, noting

that our approach generalizes straightforwardly to multigraphs (i.e., allowing multiple edges between pairs of nodes in $E$ and $E'$) and maintains its performance guarantees as long as the maximum degree of each node is constant (that is, the size of the action space is $O(1)$).

Informally, an algorithm that discovers useful hierarchies would satisfy the following desiderata:

1. Favor smaller clusters.

2. Favor a small number of clusters.

3. Favor dense connectivity within clusters.

4. Favor sparse connectivity between clusters [21], with the exception of the bridges that connect clusters.

Intuitively, having too few (for example, one) or too many clusters (for example, each state is its own cluster) creates a degenerate hierarchy that reduces the planning problem to the flat scenario, and hence medium-sized clusters are best (desiderata 1 and 2). Additionally, the hierarchy ignores transitions across clusters, which could lead to suboptimal paths generated by the hierarchical planner. It is therefore best to minimize the number of cross-cluster transitions (desiderata 3 and 4). The exception is bridges, which correspond to the links between clusters.

These desiderata can be formalized as a generative model for hierarchies and environments (Fig 3B):

$$c \sim CRP(\alpha) \qquad \text{cluster assignments} \tag{1}$$

$$p' \sim Beta(1, 1) \qquad \text{density in } H \tag{2}$$

$$Pr[(w, z) \in E'] = p' \qquad \text{edges in } H \tag{3}$$

$$Pr[b_{w,z} = (u, v) \mid (w, z) \in E', c_u = w, c_v = z] = \frac{1}{n_w n_z} \qquad \text{bridges} \tag{4}$$

$$p \sim Beta(1, 1) \qquad \text{within-cluster density in } G \tag{5}$$

$$q \sim Beta(1, 1) \qquad \text{cross-cluster density penalty in } G \tag{6}$$

$$Pr[(u, v) \in E \mid c_u = c_v] = p \qquad \text{within-cluster edges in } G \tag{7}$$

$$Pr[(u, v) \in E \mid c_u \neq c_v, b_{c_u, c_v} \neq (u, v)] = pq \qquad \text{cross-cluster edges in } G \tag{8}$$

$$Pr[(u, v) \in E \mid b_{c_u, c_v} = (u, v)] = 1 \qquad \text{bridge edges in } G \tag{9}$$

Where $n_w = |\{u : c_u = w\}|$ is the size of cluster $w$ and CRP is the Chinese restaurant process, a nonparametric prior for clusterings [22]. We additionally impose the constraint that no cluster is disconnected, that is, the induced subgraph for each cluster forms a single connected component (see Methods).

Eq 1 fulfills desiderata 1 and 2, with the concentration parameter $\alpha$ determining the trade-off between the two: lower values of $\alpha$ favor few, larger clusters, while higher values of $\alpha$ favor more, smaller clusters. Eqs 2 and 3 generate the high-level graph $H$, with higher

values of $p'$ making $H$ more densely connected. Eq 4 generates the bridges by connecting a random pair of nodes $(u, v)$ for each pair of connected clusters $(w, z)$. Eqs 5 and 7 fulfill desideratum 3 by generating the low-level edges in $G$ within each cluster, with higher values of $p$ resulting in dense within-cluster connectivity. Eqs 6 and 8 fulfill desideratum 4 by generating the low-level edges across clusters, with higher values of $q$ resulting in more cross-cluster edges. Note that $pq < p$ and hence the density of cross-cluster edges will always be lower than the density of within-cluster edges. Finally, Eq 9 ensures that each bridge edge always exists.

This generative model captures the agent's subjective belief about the generative process that gave rise to the environment and the observations. This belief could itself have been acquired from experience or be evolutionarily hardwired. The assumed generative process defines a joint probability distribution $P(G, H) = P(G|H)P(H)$ over the observable graph $G$ and the hidden hierarchical structure $H$ that generated it. Importantly, the generative process is biased to favor graphs $G$ with a particular "clustered" structure. In order to discover the underlying hierarchy $H$ and to use it to plan efficiently, the agent must "invert" the generative model and infer $H$ based on $G$.

Formally, hierarchy discovery can be framed as performing Bayesian inference using the posterior probability distribution over hierarchies $P(H|G)$:

$$
\begin{aligned}
P(H|G) \quad &= \frac{P(G|H)P(H)}{P(G)} \\
&\propto P(E|c, b, p, q)P(p)P(q)P(b|E', c)P(E'|p')P(p')P(c)
\end{aligned}
\tag{10}
$$

This predicts that neighboring states which are more densely connected will tend to be clustered together. We assess this prediction in simulations one through five.

Note that while our generative model is motivated by normative considerations, it does not comprise a formal analysis of optimality. Such analyses have been reported previously [8] and fail to capture all the effects reported here (Table 2). Furthermore, any such analysis would require, at minimum, the assumption of a probability distribution over graphs, tasks, and rewards. Our generative model is equivalent to precisely such a probability distribution, from which we can directly derive a recognition model by framing hierarchy discovery as inference over that probability distribution, thus obviating the need for any additional assumptions.

**Task distribution.** Previous studies have demonstrated that people discover hierarchies based on topological structure (simulations one through five). However, other factors may also play a role. In particular, the distribution of tasks that an agent faces in the environment may make some hierarchies less suitable than others [11], independently of the graph topology. For example, if the agent has to frequently navigate from state $s$ to state $g$ in the graph $G$ in Fig 3A, then the current hierarchy $H$ would clearly be a poor choice, even if it captures the topological community structure of $G$ well. Since hierarchical planning is always optimal within a cluster, one way to accommodate tasks is to cluster together states that frequently co-occur in the same task.

Casting hierarchy discovery as hidden state inference allows us to formalize this intuition with a straightforward extesion to our model. Following previous authors [2, 23], we assume the agent faces a sequence of tasks in $G$, where each task is to navigate from a starting state $s \in V$ to a goal state $g \in V$. We assume the agent prefers shorter routes. Defining $tasks = \{task_t\}$

and $task_t = (s_t, g_t)$, we can extend the generative model with (Fig 3C):

$$p'' \sim Beta(1, 1) \qquad \text{cross-cluster task penalty} \tag{11}$$

$$Pr[s_t = u] \;\; = \frac{1}{N} \qquad \text{starting states} \tag{12}$$

$$Pr[g_t = v \mid s_t = u] \;\; \propto \begin{cases} 1 & \text{if } c_u = c_v \\ p'' & \text{otherwise} \end{cases} \quad \text{goal states} \tag{13}$$

where $c_u$ and $c_v$ are the cluster assignments of states $u$ and $v$, and $N = |V|$ is the total number of states. Eq 12 expresses the agent's belief that a task can start randomly in any state. Eq 13 expresses the belief that tasks are less likely to have goal states in a different cluster, with $p''$ controlling exactly how much less likely that is.

If we denote the observable data as $D = (tasks, G)$, the posterior becomes:

$$\begin{aligned} P(H|D) \;\; &\propto P(D|H)P(H) \\ &= P(tasks|G, H)P(G|H)P(H) \\ &= \left[ \prod_t P(g_t|s_t, p'', G, H)P(s_t|G, H) \right] P(p'')P(G|H)P(H), \end{aligned} \tag{14}$$

where the last two terms are the same as in Eq 10.

The model will thus favor hierarchies that cluster together states which frequently co-occur in same tasks. This predicts that, in the absence of community structure, hierarchical planning will occur over clusters delineated by task start and goal states. This is a key novel prediction of our model which we assess in experiments one through five.

**Reward distribution.** Besides topology and tasks, another factor that may play a role in hierarchy discovery is the distribution of rewards in the environment. In accordance with RL, we assume that each state delivers stochastic rewards and the agent aims to maximize the total reward [13]. Hierarchy discovery might account for rewards by clustering together states that deliver similar rewards. This is consistent with the tendency for humans to cluster based on perceptual features [2] and would be rational in an environment with autocorrelation in the reward structure [24–26]. Note that from the perspective of planning alone, rewards or mere perceptual similarity should be irrelevant. This highlights one way in which our model departs from formal notions of optimality. Adhering to the idea of hidden states in structure learning [10], it treats state chunks akin to latent causes which generate similar observations, so it can use those observations to infer the unobservable state chunks.

We can incorporate this intuition into the generative model by positing that states in the same cluster deliver similar rewards (Fig 3D):

$$\theta_w \sim \mathcal{N}(\bar\theta, \sigma_\theta^2) \qquad \text{average cluster rewards} \tag{15}$$

$$\mu_v \sim \mathcal{N}(\theta_{c_v}, \sigma_\mu^2) \qquad \text{average state rewards} \tag{16}$$

$$r_{v,t} \sim \mathcal{N}(\mu_v, \sigma_r^2) \qquad \text{rewards} \tag{17}$$

where $w \in V'$ and $v \in V$. $\bar\theta$ is the average reward of all states, $\theta_w$ is the average reward of states in cluster $w$, $\mu_v$ is the average reward of state $v$, $r_{v,t}$ is the actual reward delivered by state $v$ at time $t$, and $\sigma_r^2$ is the variance of that reward.

The observable data thus becomes $D = (r, tasks, G)$ and the posterior can be computed as:

$$
\begin{aligned}
P(H|D) \quad &\propto P(D|H)P(H) \\
&= P(r|tasks, G, H)P(tasks|G, H)P(G|H)P(H) \\
&= \left[ \prod_v \left[ \prod_t P(r_{v,t}|\mu_v) \right] P(\mu_v|\theta_{c_v}, H) \right] \left[ \prod_w P(\theta_w|H) \right] P(tasks|G, H)P(G|H)P(H)
\end{aligned}
\tag{18}
$$

The model will thus favor hierarchies that cluster together states which deliver similar rewards. This predicts a certain pattern of reward generalization, with states inheriting the rewards of other states in the same cluster. Importantly, this implies that boundaries in the reward landscape will induce clusters that in turn will influence planning. This is another key novel prediction of our model which we assess in experiments six and seven.

**Inference.** We approximated Bayesian inference using Markov chain Monte Carlo (MCMC) sampling (see Methods) to draw samples approximately distributed according to $P(H|D)$. We simulate each participant by drawing a single hierarchy $H$ (the sampled hierarchy) from the posterior and then using it to make decisions. This is equivalent to assuming participants perform probability matching in the space of hierarchies.

In all simulations, we assume perfect (lossless) memory for the observations $D = (r, tasks, G)$. While the process of learning the graph structure and the task and reward distributions is interesting in its own right, our focus is on inferring the hidden hierarchy $H$. We thus aim to develop a computational-level theory (in the Marrian sense) [27] of hierarchy discovery that remains agnostic of the particular algorithmic and implementational details but rather instantiates an entire class of process models that could approximate the ideal Bayesian observer.

**Choices.** For simulations one through five, we simulate choices based on $H$ using linking assumptions analogous to those used by the authors of the original studies. For experiments one through five and experiment seven, we simulate hierarchical planning based on $H$ using HBFS. For experiment six, we assume participants prefer the state with the highest expected reward. For experiment eight (see S1 Appendix), we assume participants prefer to reduce the entropy of the posterior as much as possible. In order to account for choice stochasticity, for each decision, we simulate the appropriate choice as dictated by the model with probability $\epsilon$, or choose randomly with probability $1 - \epsilon$. We picked all model parameters by hand based on simulations one through five and based on the design for experiments six and seven. We used the same parameters throughout all simulations and experiments (Table 1).

## Simulations

Our hierarchy discovery framework can account for a wide range of behavioral effects which present difficulty for existing models (Table 2). To assess its basic predictions, we simulate a

**Table 1. Model parameter settings.** These were held constant across all simulations and experiments.

| Parameter | Range | Role | Value |
|---|---|---|---|
| $\alpha$ | $[0, +\infty)$ | CRP concentration parameter: larger values favor more clusters | 1 |
| $\epsilon$ | $[0, 1]$ | choice stochasticity: larger values lead to more deterministic choices | 0.6 |
| # samples | $[1, +\infty)$ | number of MCMC iterations per simulated participant | 10000 |
| $\bar{\theta}$ | $(-\infty, +\infty)$ | average reward of entire graph | 15 |
| $\sigma_\theta$ | $(-\infty, +\infty)$ | standard deviation of average cluster rewards around $\bar{\theta}$ | 10 |
| $\sigma_\mu$ | $(-\infty, +\infty)$ | standard deviation of average state rewards | 10 |
| $\sigma_r$ | $(-\infty, +\infty)$ | standard deviation of state rewards | 5 |

**Table 2. Model comparison.** Summary of which results could potentially be accounted for by alternative models and which results rule out certain models.

| Model / Result | Efficient state space modularization [34] | Optimal behavioral hierarchy [8] | Temporal community structure [7] |
|---|---|---|---|
| Simulation one: bottleneck transitions | YES | YES | YES |
| Simulation two: bottleneck states | YES | YES | YES |
| Simulation three: hierarchical planning | YES | YES | NO (representations not suitable for planning) |
| Simulation four: shorter hierarchical paths | YES | YES | NO (representations not suitable for planning) |
| Simulation five: cross-cluster jumps | YES | YES | YES |
| Experiment one: task distributions | YES | NO (fixed task distribution) | NO (no task distribution) |
| Experiment two: task distributions and suboptimal planning | YES | NO (fixed task distribution) | NO (no task distribution) |
| Experiment three: learning effects | NO (no uncertainty) | NO (fixed task distribution) | NO (no task distribution) |
| Experiment four: perfect information | YES | NO (fixed task distribution) | NO (no task distribution) |
| Experiment five: perfect information and suboptimal planning | YES | NO (fixed task distribution) | NO (no task distribution) |
| Experiment six: reward generalization | NO (no reward distribution) | NO (no reward distribution) | NO (no reward distribution) |
| Experiment seven: rewards and planning | NO (no reward distribution) | NO (no reward distribution) | NO (no reward distribution) |
| Experiment eight (S1 Appendix): uncertainty and active learning | NO (no uncertainty) | NO (no uncertainty) | NO (no uncertainty) |

series of behavioral experiments reported in the literature. Our primary focus is on validating the foundational assumptions behind our model without committing to any particular parameterization. We therefore use the same set of generic handpicked parameters across all of our simulations (Table 1) and focus primarily on reliable qualitative effects that are independent of the particular choice of parameters.

**Simulation one: Bottleneck transitions.** Schapiro et al. [7] demonstrated that people can detect transitions between states belonging to different clusters in a graph with a particular topological structure, such that nodes in certain parts of the graph are more densely connected with each other than with other nodes. This type of topology is also referred to as *community structure* (Fig 4A), with the clusters built into the graph topology referred to as *communities*. Thirty participants viewed sequences of stimuli representing random walks or Hamiltonian

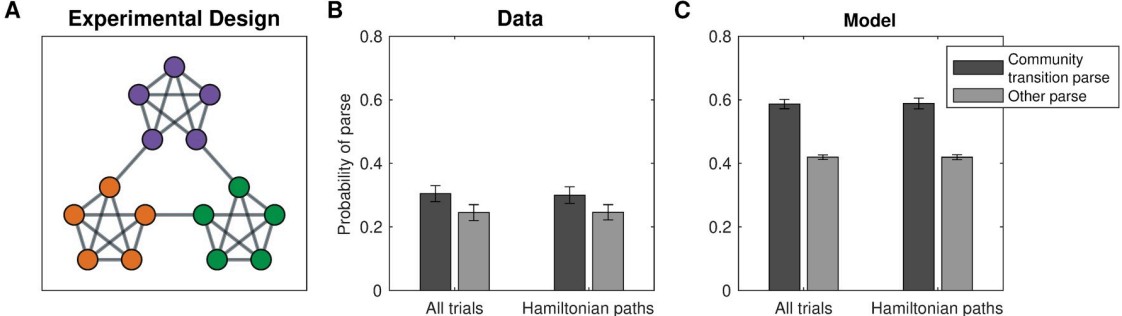

**Fig 4. Detecting transitions between communities.** A. Graph from Schapiro et al. [7]. Colors visualize the communities of states. Participants never saw the graph or received hints of the community structure. B. Results from Schapiro et al. [7], experiment 1, showing that participants were more likely to parse the graph along community boundaries. Participants indicated transitions across communities as "natural breaking points" more often than transitions within communities. Error bars are s.e.m. (30 participants). C. Results from simulations showing that hierarchy inference using our model is also more likely to parse the graph along community boundaries. Error bars are s.e.m. (30 simulations).

paths in the graph. Participants were instructed to press a key whenever they perceived a natural breaking point in the sequence. The authors found that participants pressed significantly more for cross-community transitions than for within-community transitions (Fig 4B). Participants did this despite never seeing a bird's-eye view of the graph or receiving any hints of the community structure.

Following experiment 1 from Schapiro et al. [7], for each simulated participant, we sampled a hierarchy $H$ based on the graph $G$ and performed 18 random walks of length 15 initiated at random nodes, and 18 random Hamiltonian paths. For simplicity, we simulated key presses deterministically. In particular, we counted a transition from node $u$ to node $v$ as a natural breaking point if and only if the nodes belonged to different clusters in the inferred hierarchy $H$, that is, if $c_u \neq c_v$, where $c$ are the cluster assignments in $H$. This recapitulated the empirical results (Fig 4C; random walks: $t(58) = 7.35$, $p < 10^{-9}$; Hamiltonian paths: $t(58) = 7.32$, $p < 10^{-9}$, two sample, two-tailed t-tests).

The dense connectivity within communities and sparse connectivity across communities drives the posterior to favor hierarchies with cluster assignments similar to the true underlying community structure. This arises due to Eqs 7 and 8 in the generative model, corresponding to desiderata 3 and 4, respectively. This posits that, during the generative process, edges across clusters are less likely than edges within clusters, resulting in a posterior distribution that penalizes alternative hierarchies in which many edges end up connecting nodes in different communities.

**Simulation two: Bottleneck states.** Solway et al. [8] performed several experiments demonstrating that people spontaneously discover graph decompositions that fulfill certain formal criteria of optimality. Similarly to Solway et al. [7], in their first experiment, forty participants were trained on a graph with community structure (Fig 5A). As before, participants never saw the full graph or were made aware of its community structure but instead had to to rely solely on transitions between nodes. Participants were then asked to designate a single node in the graph as a "bus stop", which they were told would reduce navigation costs in a subsequent part of the experiment. Participants preferentially picked the two bottleneck nodes on the edge that connects the two communities (Fig 5B), which are the optimal subgoal locations under these constraints. This suggests that participants were able to infer the graph topology based on adjacency information only, and to decompose it in an optimal way.

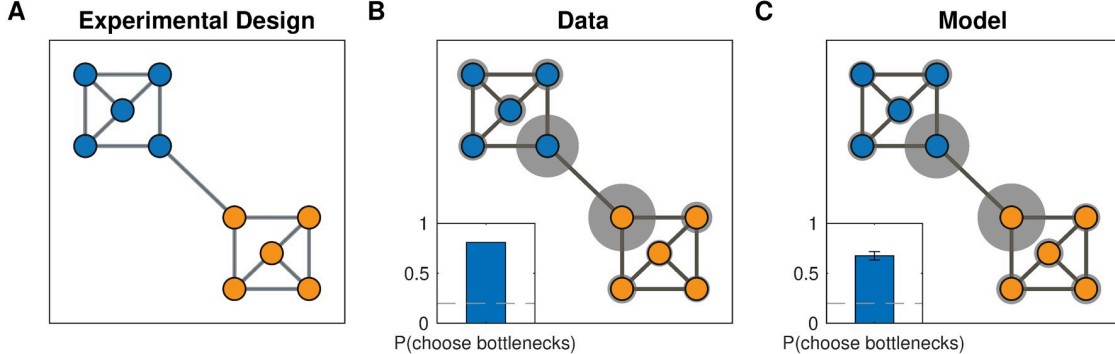

**Fig 5. Detecting bottlenecks states.** A. Graph from Solway et al. [8], experiment 1, with colors indicating the optimal decomposition according to their analysis. B. Results from Solway et al. [8], experiment 1, showing that people are more likely to select the bottleneck nodes as bus stop locations. Gray circles indicate the relative proportion of times the corresponding node was chosen. Inset, proportion of times either bottleneck node was chosen. Dashed line is chance (40 participants). C. Results from simulations showing that our model is also more likely to pick the bottleneck nodes since they are more likely to end up as endpoints of a bridge. Notation as in B. Inset error bars are s.e.m (40 simulations).

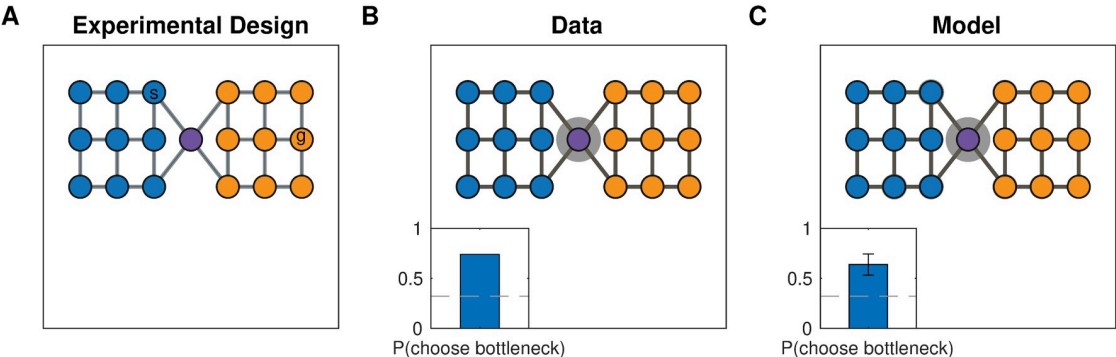

**Fig 6. Planning transitions across communities first.** A. Graph from Solway et al. [8], experiment 2, with colors indicating the optimal decomposition according to their analysis. The nodes labeled *s* and *g* indicate an example start node and goal node, respectively. B. Results from Solway et al. [8], experiment 2, showing that people are more likely to think of bottlenecks states first when they plan a path between states in different communities. Notation as in Fig 5B (10 participants). C. Results from our simulation demonstrating that our model also shows the same preference. Using the hierarchy identified by our model, the hierarchical planner is more likely to consider the bottleneck state first, since it is more likely to end up as the endpoint of a bridge connecting the two clusters. Error bars are s.e.m (10 simulations).

As in simulation one, for each participant we sampled a hierarchy *H* based on the graph *G* used in the experiment. Since participants were asked to identify three candidate bus stops, we randomly sampled three nodes among all nodes that belonged to bridges in *H*, i.e. $\{u: b_{w,z} = (u, v)$, for some $(w, z) \in E'$ and $u \in V\}$, where the *b* and *E'* are the bridges and edges in the sampled hierarchy, respectively. This replicated the empirical result (Fig 5C; 65% of choices, $p < 10^{-20}$, right-tailed binomial test), with most simulated participants inferring hierarchies respecting the underlying community structure. Similarly to simulation one, this was due to the higher connectivity within communities than across communities.

**Simulation three: Hierarchical planning.** In their second experiment, Solway et al. [8] trained 10 participants to navigate between pairs of nodes in a different graph (Fig 6A). On some trials, participants were asked to indicate a single node that lies on the shortest path between two nodes in different communities. They found that participants overwhelmingly selected the bottleneck node between the two communities (Fig 6B), suggesting that they not only discovered the underlying community structure, but also leveraged that structure to plan hierarchically, "thinking first" of the high-level transitions between clusters of states.

As in simulations one and two, we sampled *H* based on the graph *G* for each participant. We then used the hierarchical planner to find 50 hierarchical paths between random start locations in the left community and random goal locations in the right community. For each such path, we counted the first node that belongs to a bridge as the response on the corresponding trial, since this is the first node considered by the planner and is therefore the closest approximation to what a participant would think of first (see definition of HBFS in the Methods). This replicated the empirical results, with a strong tendency for the bottleneck location to be selected (Fig 6C; 60% of choices, $p < 0.001$, right-tailed Monte Carlo test). The discovered hierarchies resembled the underlying community structure for the same reason as in simulations one and two, resulting in the bottleneck node frequently becoming part of a bridge that all paths between the two communities would pass through.

**Simulation four: Shorter hierarchical paths.** In their final experiment, Solway et al. [8] demonstrated hierarchical decomposition and planning in the Towers of Hanoi task, which can be represented by a graph (Fig 7A) in which each node is a particular game state and each edge corresponds to move that transitions from one game state to another. They leveraged the

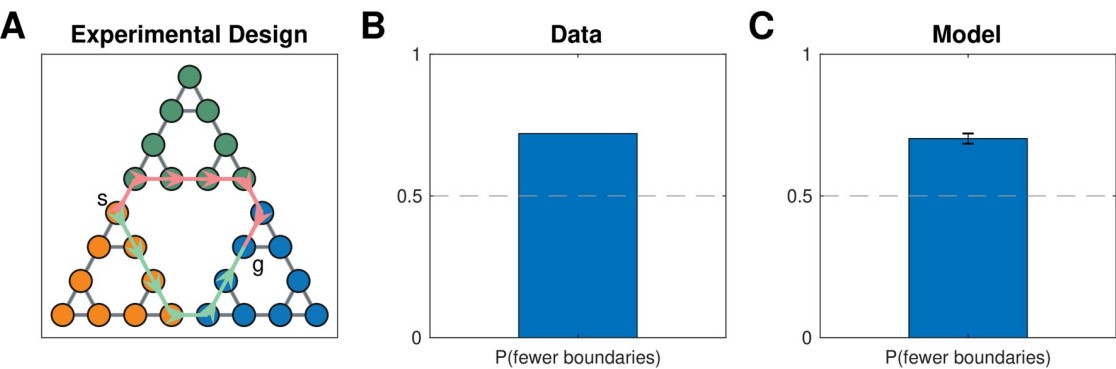

**Fig 7. Preferring paths with fewer community boundaries.** A. Graph representing the Towers of Hanoi task used in Solway et al. [8], experiment 4. Vertices represent game states, edges represent moves that transition between game states. The start and goal states (*s, g*) show an example of the kinds of tasks used in the experiment. Colored arrows denote the two shortest paths that could accomplish the given task, with the red path passing through two community boundaries and the green path passing through a single community boundary. B. Results from Solway et al. [8], experiment 4, showing that participants preferred the path with fewer communities, or equivalently, the path that crosses fewer community boundaries. Bar graph shows fraction of participants (35 participants). Dashed line is chance. C. Results from simulations showing that our model also exhibits the same preference. Bar graph shows the fraction of simulations that chose the path with fewer community boundaries. Error bar is s.e.m. (35 simulations).

fact that there are two different shortest paths between some pairs of states (for example, the start and goal states in Fig 7A), but those paths cross a different number of community boundaries as defined by their optimal decomposition analysis. Hierarchical planning predicts that participants will prefer the path which crosses fewer community boundaries, and this is indeed what the authors found (Fig 7B).

After choosing a hierarchy *H* for each simulated participant as in the previous simulations, we then used the hierarchical planner to find paths between the six pairs of states that satisfy the desired criteria. In accordance with the data, this resulted in the path with fewer community boundaries being selected more frequently (Fig 7C; 71.4% of choices, $p < 10^{-10}$, right-tailed binomial test). Similarly to the previous simulations, the model tended to carve up the graph along community boundaries. Since the planner first plans in the high-level graph, it prefers the path with the fewest clusters, and hence with the fewest cluster boundaries.

**Simulation five: Cross-cluster jumps.** In our final simulation, we considered a study performed by Lynn et al. [12], in which participants experienced a random walk along the graph shown in Fig 8A. This was a self-paced serial reaction time task in which participants had to press a different key combination for each node. A small subset of transitions violated the graph structure and instead "teleported" the participant to a node that is not connected to the current node. Importantly, there were two types of violations: short violations of topological distance 2 and long violations of topological distance 3 or 4. The authors found that participants were slower to respond to longer than to shorter violations, suggesting that participants had inferred the large scale structure of the graph.

While this is not a planning task, we assumed that reaction times (RTs) for cross-cluster transitions would be slower for the same reason as in experiment three: a cross-cluster transition requires planning in the high-level graph, imposing a kind of context switch cost that would slow RTs.

Like in the previous simulations, we sampled *H* for each participant and then simulated a random walk along the graph *G*, with occasional violations as described in Lynn et al. [12]. In order to model RTs, we assumed a bimodal distribution of RTs, with fast RTs for transitions within clusters and slow RTs for transitions across clusters, consistent with the notion that cross-cluster transitions are more surprising. Instead of actually simulating RTs, we simply

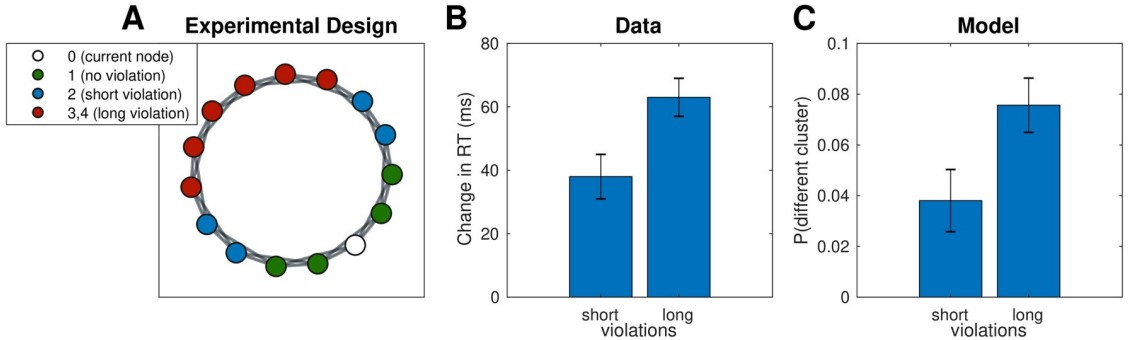

**Fig 8. Slower reactions to cross-cluster transitions.** A. Graph used in Lynn et al. [12]. Each node (white) is connected to its neighboring nodes and their neighbors (green). Blue nodes are 2 transitions away from the white node, while red nodes are 3 or 4 transitions away. B. Results from Lynn et al. [12] showing that, on the test trial, participants were slower to respond to long violations than to short violations. Change in RT is computed with respect to average RT for no-violation transitions. Error bars are s.e.m (78 participants). RT, reaction time. C. Results from simulations showing that long violations are more likely to end up in a different cluster, which would elicit a greater surprise and hence a slower RT, similar to crossing a cluster boundary.

counted the number of cross-cluster transitions during the random walk. This revealed a greater number of cross-cluster transitions for long violations than for short violations, consistent with the data (Fig 8B; $t(154) = 4.50$, $p = 0.00001$, two-sample t-test). This occurs because nearby nodes are more likely to be clustered together, and hence violations of greater topological distance increase the likelihood that the destination node would be in a different cluster than the starting node, resulting in a greater surprise and a slower RT.

## Experiments

Framing hierarchy discovery as a particular kind of hidden state inference allows us to straightforwardly extend our framework to account for the distribution of tasks and rewards in the environment. The best of our knowledge, the effect of tasks and rewards on hierarchy discovery has not been systematically investigated in prior empirical work. In particular, our model predicts that people will cluster together adjacent states that regularly co-occur in the same task (Eqs 12 and 13) as well as adjacent states that deliver similar rewards (Eqs 15, 16 and 17). Note that these novel predictions are not accounted for by any existing models (Table 2). Some of these predictions are counterintuitive—for example, is not clear why states associated with the same reward should be relevant from the perspective of planning. Understanding state chunks as hidden states which deliver observations from the same distribution—similar in spirit to latent cause models in structure learning [10, 28] – sheds insight into these phenomena, which we assess in a series of eight behavioral experiments. Each experiment aims to verify a unique prediction of our framework, or to provide a nuance or address a potential concern related to a previous experiment. As such, we believe that no experiment stands on its own and that our empirical results should be considered together, providing convergent evidence that hierarchy discovery is a form of structure learning in which the inferred state chunks are shaped by tasks and rewards in addition to environment topology.

## Experiment one: Task distributions

In our first experiment, we sought to validate the prediction of our model that clusters could be induced by the distribution of tasks alone, even when the graph topology does not favor any particular clustering. In particular, our model predicts that states which frequently co-occur in

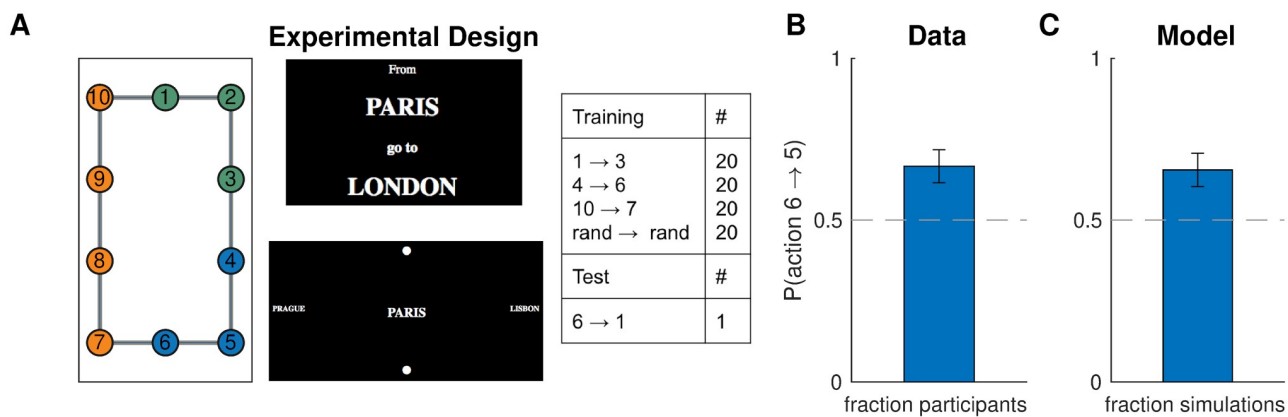

**Fig 9. Hierarchy discovery is sensitive to the task distribution.** A. (Left) graph used in experiment one with no topological community structure. Colors represent clusters favored by the training protocol (right). Numbers serve as node identifiers and were not shown to participants. "Rand" denotes a node that is randomly chosen on each trial. (middle) trial instruction (top) and screenshot from the starting state (bottom). B. Results from experiment one showing that, on the test trial, participants were more likely to go to state 5 than to state 7, indicating a preference for the route with fewer cluster boundaries. Dashed line is chance. Error bars are s.e.m. (87 participants) C. Results from simulations showing that our model also preferred the transition to state 5. Notation as in B.

the same task should be clustered together, since the hierarchical planner is optimal within clusters.

**Participants.** We recruited 87 participants (28 female) from Amazon Mechanical Turk (MTurk). All participants received informed consent and were paid $3 for their participation. We reasoned that paying participants a fixed amount would incentivize them to complete the experiment in the least amount of time possible, which entails balancing path length with planning time in a way that is characteristic of many real-world tasks. The experiment took 17 minutes on average. All experiments were approved by the Harvard Institutional Review Board.

**Design.** We asked participants to navigate between pairs of nodes ("subway stops") in a 10-node graph (the "subway network", Fig 9A, left). The training trials (Fig 9A, right) were designed to promote a particular hierarchy: the task to navigate from node 1 node 3 would favor clustering nodes 1,2,3 together; the task to navigate from node 4 to node 6 would favor clustering nodes 4,5,6 together; and the task to navigate from node 10 to node 7 would favor clustering nodes 7,8,9,10 together (Fig 9A, left). The normative reason for this is that hierarchical planning is always optimal within a cluster. In the generative model, this is taken into account by Eq 13, which leads to a preference to cluster together start and goal states from the same task. The purpose of the tasks with random start and goal states was to encourage participants to learn a state representation for efficient planning and not simply to respond habitually.

In order to test the model prediction, after training, we asked participants to navigate from node 6 node 1. Note that the two possible paths are of the same length and a planner with a flat representation of the graph would show no preference for one path over the other. Furthermore, since there is no community structure and the graph is perfectly symmetric, any clustering strategy based on graph structure alone would not predict a preference. Conversely, our model predicts that participants will tend to choose the path through node 5 since it passes through a single cluster boundary, whereas the path through node 7 passes through two cluster boundaries.

**Procedure.** The experiment was implemented as a computer-based game similar to Balaguer et al. [2] in which participants had to navigate a virtual subway network. At the start of each trial, participants saw the names of the starting station and the goal station. After 2 s, they

transitioned to the navigation phase of the trial, during which they could see the name of the current station in the middle of the screen, surrounded by the names of the four neighboring stations, one in each cardinal direction. If there was no neighboring station in a particular direction, participants saw a filled circle instead of a station name. The name of the goal station was also indicated in the top left corner of the screen as a reminder. The navigation phase began with a 3-s countdown during which participants could see the starting station and its neighbors but could not navigate. Participants were instructed to plan their route during the countdown. After the countdown, participants could navigate the subway network using the arrow keys. Transitions between stations were instantaneous. Once participants reached the goal station, they had to press the space bar to complete the trial. This was followed by a 500-ms "success" message, after which the trial ended and the instruction screen for the next trial appeared. Pressing the space bar on a non-goal station resulted in a "incorrect" message flashing on the screen. Attempting to move in a direction without a neighboring station had no effect. Following Balaguer et al. [2], stations were named after cities, with the names randomly shuffled for each participant.

The subway network corresponded to the graph in Fig 9A. In order to assign arrow keys to edges, we first embedded the graph in the Cartesian plane by assigning coordinates to each vertex, which resulted in the planar graph shown in the figure. Then we assigned the arrow keys to the corresponding cardinal directions. For each participant, we also randomly rotated the graph by 0˚, 90˚, 180˚, or 270˚. Participants performed 80 training trials (20 in each condition; Fig 9, right) in a random order. After the training trials, we showed a message saying that now the subway system was unreliable, so that some trips may randomly be interrupted midway. This was followed by the test phase, during which participants performed the test trial $6 \rightarrow 1$ and two additional test trials. In order to prevent new learning during the test phase, all test trials were interrupted immediately after the first valid keypress. The two additional test trials were not included in the analysis or any of the following experiments. We used the destination of the first transition on the test trial as our dependent measure in the analysis. We reasoned that the direction in which participants attempt to move first is along what they perceive to be the best route to the goal station. Since participants were paid a fixed fee for the whole experiment, they were incentivized to complete it as fast as possible, which can be best achieved by planning the shortest route during the 3-s countdown and then following it.

**Results and discussion.** In accordance with our model predictions, more participants moved to state 5 on the test trial, rather than to state 7 (Fig 9B; 58 out of 87, $p = 0.003$, two-tailed binomial test). Notice that this could not be explained by habitual responding: while participants may have learned action chunks that solve the corresponding tasks (for example, pressing right and down from state 1 to state 3), the actions from state 6 to its neighboring states were never reinforced as part of a stimulus-response association or as part of a longer action chunk. In particular, state 6 is never a starting state or an intermediary state, except possibly in the random tasks, in which both directions have equal probability of being reinforced. The effect cannot be explained by state familiarity either, since participants experienced states 5 and 7 equally often, on average. Finally, it is worth noting that a model-free RL account would make the opposite prediction, since state 7 would on average have a higher value than state 5, as it gets rewarded directly.

This result was consistent with the model predictions (Fig 9C; 57 out of 87, $p = 0.005$, two-tailed binomial test), suggesting that people form hierarchical representations for planning in a way that tends to cluster together nodes that co-occur in the same tasks.

In order to rule out the possibility that the preference for the $6 \rightarrow 5$ transition on the test trial was simply due the frequent $5 \rightarrow 6$ transitions during training, which might have somehow reinforced the reverse action, we performed a logistic regression of test trial choices on

the total number of $6 \rightarrow 5$, $5 \rightarrow 6$, $6 \rightarrow 7$, and $7 \rightarrow 6$ transitions during training. This did not show a significant effect for any of the transitions ($F(1, 83) < 2.8$, $p > 0.09$ for all coefficients), thus disproving such a "flat", non-hierarchical associative account.

## Experiment two: Task distributions and suboptimal planning

Since both paths on the test trial in the previous experiment were of equal length, the bias that participants developed would make no difference for their performance on that task. Next we asked whether such a bias would occur even if it might lead to suboptimal planning, as the hierarchical planner would predict. For example, most of us have had the experience of navigating between two locations through other, more familiar location, even though a shorter but less familiar route might exist (see also [29]).

**Participants.** We recruited 241 participants (112 female) from MTurk. Of those, 78 were assigned to the "bad" clusters condition, 87 were assigned to the "control" condition, and 76 were assigned to the "good" clusters condition. Participants were paid $2.50 for their participation. The experiment took 15 minutes on average.

**Design and procedure.** We used the same paradigm as experiment one, with the only difference that node 10 was removed from the graph (Fig 10A, left). Now the analogous training regime (Fig 10A, right) would promote "bad" clusters that lead to suboptimal planning on the test trial. We also performed a control version of the experiment with different participants using random training tasks only, as well as a "good" condition with a third group of participants that promotes clusters that lead to optimal planning on the test trial (Fig 10A).

**Results and discussion.** On the test trial, participants preferred the suboptimal move from 6 to 5 in the "bad" clusters condition (Fig 10B; 53 out of 78, $p = 0.002$, two-tailed binomial test), significantly more than the control condition ($\chi^2(1, 165) = 6.52$, $p = 0.01$, chi-square test of independence) and the "good" condition ($\chi^2(1, 154) = 10.4$, $p = 0.001$). This was in

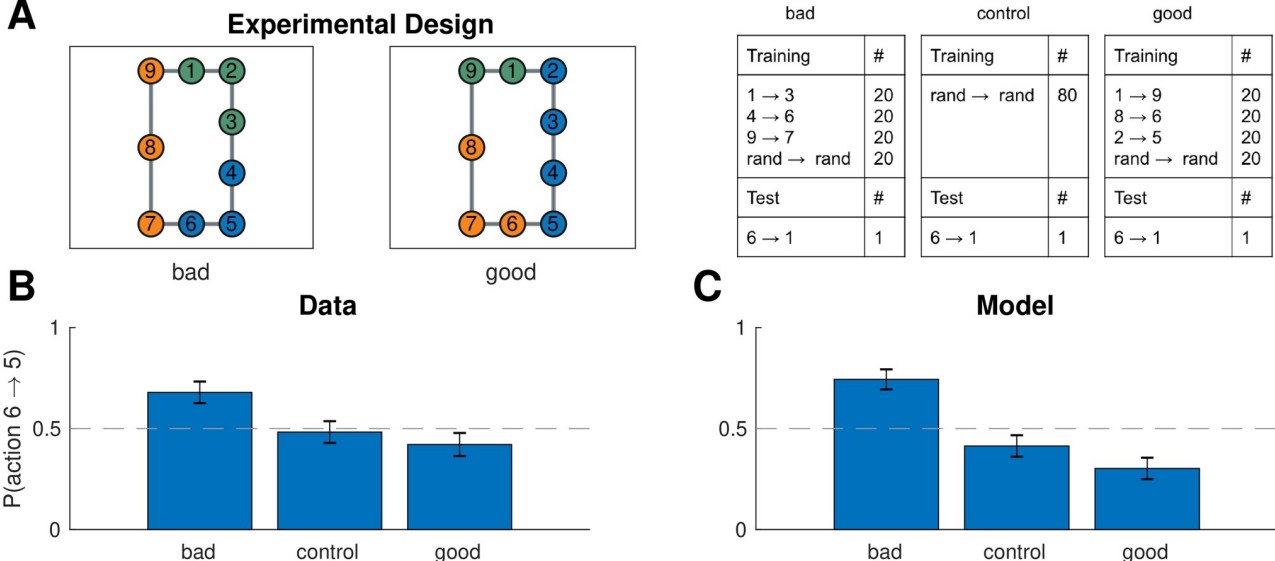

**Fig 10. Different task distributions can induce different hierarchies in the same graph.** A. (Left) graph used in experiment two with colors representing clusters favored by the training protocol in the "bad" (left) and "good" (middle) condition. (Right) training and test protocols for all three conditions. B. Results from experiment two showing that, on the test trial, participants were more likely to go to state 5 than to state 7 in the bad condition, leading to the suboptimal route. The effect was not present in the control condition or in the good condition. Dashed line is chance. Error bars are s.e.m. (78, 87, and 76 participants, respectively). C. Results from simulations showing that our model exhibited the same pattern. Notation as in B.

accordance with the model predictions (Fig 10C; 58 out of 78 simulated participants, $p = 0.00002$, in the bad condition; $\chi^2(1, 165) = 18.2$, $p = 0.00002$ for bad vs. control condition; $\chi^2(1, 154) = 30.0$, $p < 10^{-7}$ for bad vs. good condition). This suggests that participants formed clusters based on the distribution of tasks and used these clusters for hierarchical planning, even when that was suboptimal on the particular test task.

Note that the model showed a slight preference for the $6 \rightarrow 7$ transition in the control condition, which did not reach significance. This occurs because there are fewer nodes along that path and, in the presence of random clusters induced by the random tasks, there will be, on average, fewer cluster boundaries along that shorter path. Participants exhibited a similar qualitative pattern, which however was not significant. Somewhat surprisingly, there was also no significant difference between the control condition and the good condition in the data ($\chi^2(1, 163) = 0.6$, $p = 0.5$) and the model ($\chi^2(1, 163) = 2.2$, $p = 0.14$), although both showed a slightly greater preference for the $6 \rightarrow 7$ transition compared to the control condition. This suggests that in this particular scenario, the "good" clusters have a relatively weaker effect.

### Experiment three: Learning effects

While the experiments so far demonstrate the key predictions of our model, they only test asymptotic behavior and as such do not assess how beliefs about hierarchy evolve over the course of learning. Further, these results could conceivably be explained by simpler, non-hierarchical accounts, such as a bidirectional association forming between states 5 and 6 due to the frequent transition from 5 to 6 during training. While we addressed this in our previous analysis, in our next experiment we sought to definitively rule out such "flat" associative explanations and to study the dynamics of learning.

**Participants.** We recruited 127 participants (54 female) from MTurk. Participants were paid $8.50 for their participation. The experiment took 47 minutes on average.

**Design.** We trained participants on the same graph experiment one, but using a different training regimen, with two stages of training and multiple probe trials (Fig 11A). The first stage of training (trials 1-103) promoted clustering states 2 and 3 together, separately from state 1, and similarly promoted clustering states 4 and 5 together, separately from state 6 (Fig 11A, first panel). In order to investigate the dynamics of hierarchy discovery, we interspersed three probe trials that tasked participants to navigate from state 6 to state 1 throughout the first stage, expecting participants' preferences to become stronger over time. Note that predictions on the probe trials are the opposite of predictions on the test trial in experiment one: the path from 6 to 1 through state 5 now crosses three cluster boundaries, whereas the path through state 7 crosses only two cluster boundaries, so participants should prefer the path through state 7. This cannot be explained by a naive associative account since state 6 is no more likely to co-occur with state 7 than with state 5.

To further assess learning, which sought to reverse this effect during the second stage of training which promoted the same clusters as experiment one (Fig 11A, second panel). Our prediction was that, in accordance with the results of experiment one, this would eliminate participants' preference for going to state 7 on the probe trials in favor of state 5, since the path through state 5 now crosses a single cluster boundary.

**Procedure.** We used the same procedure as experiment one, with the following changes. First, there was no information indicating to participants that something had changed between stages (i.e., between trials 103 and 104). Additionally, instead of having a test stage in the end, we interspersed six probe trials $6 \rightarrow 1$, spaced evenly throughout training (trials 34, 68, 103 during the first stage and trials 150, 197, 246 during the second stage). Unlike the test trials in experiment one, probe trials were indistinguishable from training trials and were not

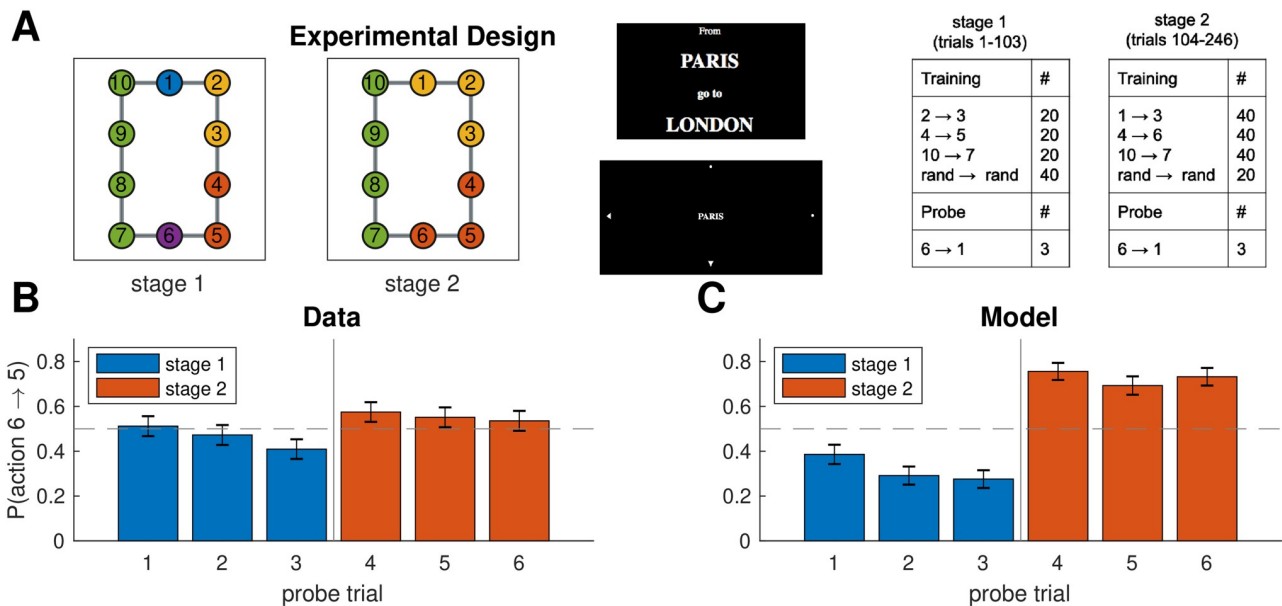

**Fig 11. Learning dynamics.** A. Experiment three used the same graph as experiment one, with main difference that training (right panel) took part in two stages that promoted different hierarchies (first and second panel), with probe trials interspersed throughout training. Notation as in Fig 9A. B. Results from experiment three showing that (1) the first stage of training makes participants more likely to go to state 7 on the probe trials, which could not be explained by a "flat" associative account, (2) this tendency appears gradually as participants accumulate more evidence, and (3) this preference is reversed during the second stage of training. Error bars are s.e.m. (127 participants). C. Results from simulations showing that our model exhibited the same learning dynamics. Notation as in B.

interrupted after the first move. Another difference from experiment one was that instead of being rotated, the graph was randomly flipped horizontally and/or vertically for each participant. Finally, unlike experiment one, participants did not see the names of adjacent stations but instead saw arrowheads indicating they could move in the corresponding direction (Fig 11A, third panel).

**Results and discussion.** Consistent with our predictions, participants showed a significant preference for moving to state 7 on the last probe trial of the first stage (Fig 11B, probe trial 3; 75 out of 127 participants, $p = 0.05$, two-tailed binomial test). This effect was modulated by the amount of training, with the smallest effect on the first probe trial and the largest effect on the third probe trial (slope = -0.27, $F(1, 379) = 3.96$, $p = 0.05$, mixed effects logistic regression with probe trials 1-3). The effect was reversed during the second stage (slope = 0.73, $F(1, 252) = 8.10$, $p = 0.005$, mixed effects logistic regression with probe trials 3-4).

These results were consistent with the model predictions (Fig 11C). The model preference for state 7 on the third probe trial (92 out of 127 simulated participants, $p < 10^{-6}$, two-tailed binomial test) is once again due to the preference of the hierarchical planner for paths with fewer state clusters. As in our empirical data, this effect became stronger over time (slope = -0.34, $F(1, 379) = 5.58$, $p = 0.02$, mixed effects logistic regression with probe trials 1-3). The reason is that, as the hierarchy inference process observes more tasks, it accumulates more evidence (i.e., more terms in the product in Eq 14) which sharpens the posterior distribution $P(H|D)$ around its mode (i.e., the most probable hierarchy, Fig 11A, first panel). This corresponds to a decrease in the uncertainty over $H$, which makes it more likely that the sampler will draw the hierarchy in Fig 11A, first panel, and plan according to it. Hence this effect of learning occurs due to the decrease in uncertainty resulting from the additional observations. Finally, like our participants, the model reversed its preference during the second stage of

training (slope = 2.13, $F(1, 252) = 53.43$, $p < 10^{-11}$, mixed effects logistic regression with probe trials 3-4). This effect occurs because tasks during the second stage of training shift the mode of the posterior (Fig 11A, second panel). Together, these results rule out simple, nonhierarchical associative accounts and demonstrate that our model can account for learning dynamics due to changes in uncertainty and changes in the mode of the posterior distribution over hierarchies.

It is worth noting that there is a significant discrepancy between the effect size predicted by the model and the effect size observed in the data, particularly with regard to the difference between the first stage and the second stage of training. We believe this is attributable to the lack of any adjacency information in this experiment, which ameliorates the potential associative confound, but by the same token renders the low-level graph $G$ more difficult to learn compared to the other experiments. This results in noisier choices, which could be modeled by adjusting the noise parameter $\epsilon$. However, since our aim was to assess robust qualitative effects that are independent of particular parameterizations, for consistency we preferred using the same set of parameters as in simulations one through five (Table 1).

One prediction of our hierarchical planner is that transitions across cluster boundaries should take longer, since they involve planning both in the high-level graph $H$ as well as the low-level graph $G$, whereas transitions within clusters only involve planning in $G$. We therefore analyzed log-transformed RTs during the first stage of training as a function of transition type using mixed effects linear regression. We classified each transition in one of three types:

- Action chunk: transitions along the shortest path for any of the training tasks ($2 \rightarrow 3, 4 \rightarrow 5, 10 \rightarrow 9, 9 \rightarrow 8, 8 \rightarrow 7$).

- State chunk: transitions along the shortest path for any of the training tasks, but in the reverse direction ($3 \rightarrow 2, 5 \rightarrow 4, 9 \rightarrow 10, 8 \rightarrow 9, 7 \rightarrow 8$).

- Boundary: all other transitions ($1 \rightarrow 2, 2 \rightarrow 1, 1 \rightarrow 10, 10 \rightarrow 1, 6 \rightarrow 7, 7 \rightarrow 6, 6 \rightarrow 5, 5 \rightarrow 6, 3 \rightarrow 4, 4 \rightarrow 3$).

We separated within-cluster transitions into action chunk transitions and state chunk transitions in order to account for the fact that actions along frequently occurring task solutions are reinforced much more frequently and are thus likely to be chunked together into stereotyped action sequences by the motor system. Consistent with this, we found that boundary transitions and state chunk transitions were both significantly slower than action chunk transitions ($F(1, 46592) = 121.00$, $p < 10^{-27}$ and $F(1, 46592) = 69.56$, $p < 10^{-16}$, respectively). In contrast, state chunk transitions are reinforced as frequently as boundary transitions, on average. Despite this, boundary transitions were slower than state chunk transitions ($F(1, 46592) = 10.68$, $p = 0.001$), as predicted by our hierarchy discovery account. Together, these results further support our hypothesis that task boundaries delineate cluster boundaries and are consistent with the notion that state abstraction drives temporal abstraction [30] – humans first decompose the environment into clusters that then constrain the chunking of actions into sequences that operate within those clusters.

## Experiment four: Perfect information

One downside of experiments one and two is that effects of memory confound hierarchy inference and planning. In particular, it is unclear that participants are able to learn and represent the full graph $G$. This is most evident in the control condition of experiment two (Fig 10B), in which our model predicts that participants should be better than chance, when in fact they are not, thus questioning whether they are learning the graph or planning efficiently in the first

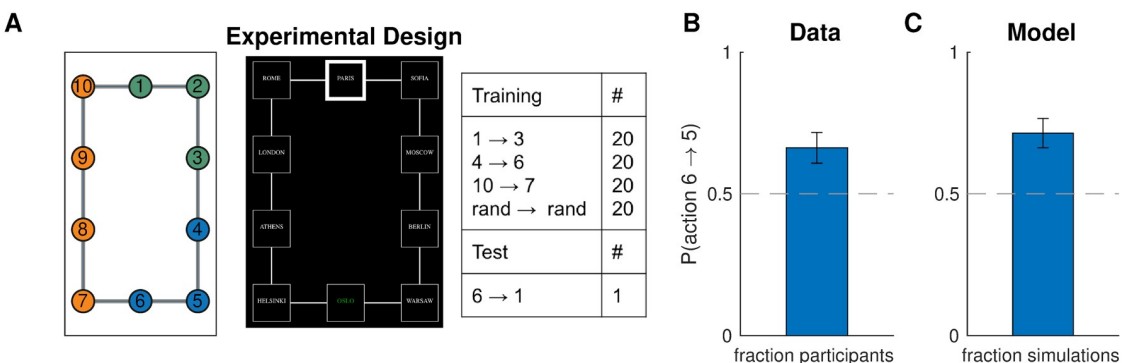

**Fig 12. Hierarchy discovery based on task distribution in fully visible graphs.** A. (Left) experiment four used the same graph as experiment one, however this time the graph was fully visible on each trial (middle). Notation as in Fig 9A. B. Results from experiment four showing that, like in experiment one, participants were more likely to go to state 5 on the test trial. Dashed line is chance. Error bars are s.e.m. (77 participants) C. Results from simulations showing that our model also preferred the transition to state 5. Notation as in B.

place. Note that this does not pose a significant challenge to our hierarchy discovery account; people's choices could still be accounted for without invoking a hierarchical planner, for example by assuming a preference to remain in the same cluster (moving to state 5) rather than crossing a cluster boundary (moving to state 7). Nevertheless, we sought to overcome this limitation by ensuring participants know the full graph *G*.

**Participants.** We recruited 77 participants (33 female) from MTurk. Participants were paid $2.00 for their participation. The experiment took 10 minutes on average.

**Design.** We used the same graph and training protocol as experiment one, except this time participants could see the whole graph at any given time (Fig 12A). This put participants on an equal footing with the hierarchy inference and hierarchy planning algorithms, both of which assume perfect knowledge of the graph *G*.

**Procedure.** The procedure was similar to experiment one, with the main difference that participants had a bird's-eye view of the entire subway network throughout the experiment (Fig 12A, middle panel). Subway stations were represented by squares connected by lines which represented the connections between stations. The current station was highlighted with a thick border and the goal station was in green font.

Since planning is significantly easier in the setting, we removed the 3-s countdown, so that participants could start navigating immediately after the 2-s instruction. Additionally, instead of rotating the map, we randomly flipped it horizontally for half of the participants. We also omitted the "unreliable trips" warning before the test phase since participants now only saw a single test trial, which was immediately interrupted after the first move. The experiment took 10 minutes on average and participants were paid $2.00.

**Results and discussion.** We found that, even with full knowledge of the graph, participants still developed a bias (Fig 12B; 51 out of 77 participants, *p* = 0.0014, right-tailed binomial test), consistent with the predictions of our model (Fig 12C; 55 out of 77, *p* = 0.0002, two-tailed binomial test). This provides strong support for our hierarchy discovery account, suggesting tasks constrain cluster inferences above and beyond constraints imposed by graph topology, which in turn constrain hierarchical planning on novel tasks.

## Experiment five: Perfect information and suboptimal planning

We next asked whether hierarchy discovery could lead to suboptimal planning even when the full graph is visible.

**Participants.** We recruited 386 participants (175 female) from MTurk. Of those, 119 were assigned to the "bad" clusters condition, 90 were assigned to the "control 1" condition, 88 were assigned to the "control 2" condition, and 89 were assigned to the "good" clusters condition. Participants were paid $2.00 for their participation. The experiment took 9 minutes on average.

**Design and procedure.** We used the same graph and training protocol as experiment two, except this time participants could see the whole graph at any given time (Fig 13A), as in experiment four. Additionally, we included two control conditions, one with 20 training trials ("control 1") and one with 80 training trials ("control 2"). The first control condition ensured participants received the same number of random tasks as the bad and good conditions, while the second control condition ensured that participants received the same total number of tasks as in the bad and good conditions. We used the same experimental procedure as experiment four.

**Results and discussion.** As in experiment two, inducing "bad" clusters still led to significantly worse performance on the test trial than either control condition (Fig 13B; $\chi^2(1, 209) = 9.35$, $p = 0.002$ for bad vs. control 1; $\chi^2(1, 207) = 7.7$, $p = 0.006$ for bad vs. control 2, chi-square test of independence). Inducing "good" clusters (89 participants) led to significantly better performance than "bad" clusters ($\chi^2(1, 208) = 9.03$, $p = 0.003$ for bad vs. good), although not significantly better than the control conditions ($\chi^2(1, 179) = 0.002$, $p = 0.96$ and $\chi^2(1, 177) = 0.05$, $p = 0.8$ for good vs. control 1 and good vs. control 2, respectively). This accords with our model predictions (Fig 13C; $\chi^2(1, 209) = 10.1$, $p = 0.002$ for bad vs. control 1; $\chi^2(1, 207) = 4.8$, $p = 0.03$ for bad vs. control 2; $\chi^2(1, 208) = 17.7$, $p = 0.00003$ for bad vs. good) and strongly suggests that people default to hierarchical planning over clusters influenced by the task distribution, even in simple, fully observable graphs. Notice that in both control conditions, participant preferred the shorter path (21 out of 90 participants, $p < 10^{-6}$ for control 1; 22 out of 88 participants, $p < 10^{-6}$ for control 2, two-tailed binomial tests), indicating that they were indeed able to plan effectively when given the full graph without tasks to bias them towards particular clusters, thus overcoming the limitation of experiment two.

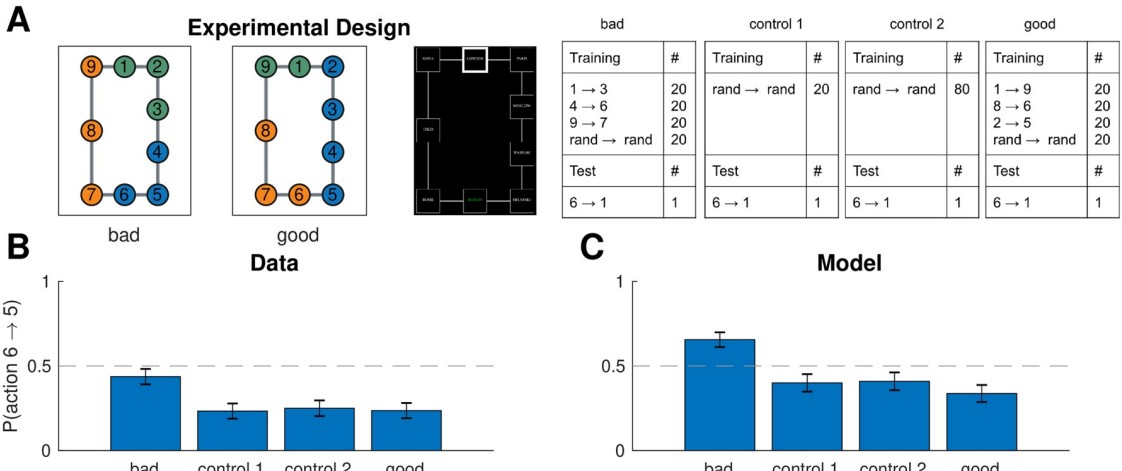

**Fig 13. Task distributions can bias hierarchical planning even in fully visible graphs.** A. (Left) experiment five used the same graph as experiment two, however this time the graph was fully visible on each trial. Notation as in Fig 10A. B. Results from experiment five showing that participants were still biased by the training tasks in the bad condition, performing worse on the test trial compared to the other conditions. Dashed line is chance. Error bars are s.e.m. (119, 90, 88, and 89 participants, respectively). C. Results from simulations showing that our model exhibited the same pattern. Notation as in B.

One notable difference between our model predictions and the empirical data is that our model predicts a preference for state 5 in the "bad" condition (78 out of 119 simulated participants, $p = 0.0009$, two-tailed binomial test), whereas participants did not show significant preference (52 out of 119 participants, $p = 0.2$, two-tailed binomial test). We believe this occurs because the task is much simpler when the graph is fully visible and participants could easily perform optimal "flat" planning, rather than having to resort to hierarchical planning. This effect could be captured straightforwardly using a mixture of BFS and HBFS for planning, rather than just HBFS.

## Experiment six: Reward generalization

In this experiment, we tested the prediction that rewards generalize within clusters. While this prediction is not unique to our model and could be accounted for by Gaussian processes over graphs [26, 31] or by the successor representation [32, 33], the idea that clusters generate similar rewards is a core assumption of our model that we sought to validate before assessing how clusters inferred based on rewards could influence planning (experiment seven).

**Participants.** We recruited 32 participants from the MIT undergraduate community. The experiment took around 3 minutes and participants were not paid for their participation.

**Design.** We showed participants the graph ("network of gold mines") in Fig 14A and told them that in the past, states delivered an average reward of 15 ("grams of gold"), but today, state 4 (green) delivered a reward of 30. We then asked participants to choose one between state 3 and state 7 ("mines to explore").

**Procedure.** Each participant was given a sheet of paper with instructions and the graph in Fig 14A, without node identifiers. The instructions were as follows:

*You work in a large gold mine that is composed of multiple individual mines and tunnels. The layout of the mines is shown in the diagram below (each circle represents a mine, and each line represents a tunnel). You are paid daily, and are paid $10 per gram of gold you found that day. You dig in exactly one mine per day, and record the amount of gold (in grams) that mine yielded that day. Over the last few months, you have discovered that, on average, each mine yields about 15 grams of gold per day. Yesterday, you dug in the blue mine in the diagram below, and got 30 grams of gold. Which of the two shaded mines will you dig in today? Please circle the mine you choose.*

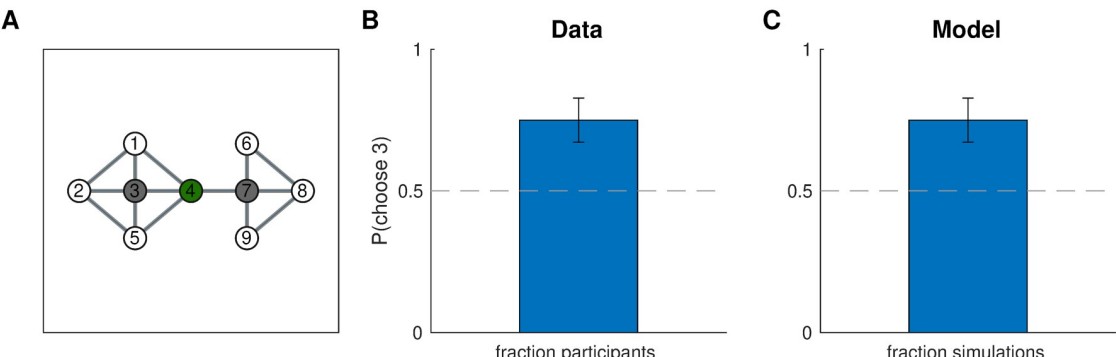

**Fig 14. Reward generalization within clusters.** A. Graph used in experiment six. Numbers indicate state identifiers and were not shown to participants. Participants were told that states deliver 15 points on average and that, on a given day, state 4 (green) delivered 30 points. They were then asked which of the two gray nodes (states 3 and 7) they would choose. B. Results from experiment six showing that participants preferred state 3, which is in the same topological cluster as state 4, suggesting they generalized the reward within the cluster. Error bars are s.e.m (32 participants). C. Results showing that the model exhibited the same pattern. Notation as in B.

Half of the participants were given a version in which the graph was flipped horizontally, i.e. the topological cluster was on the right side.

**Results and discussion.** Participants preferred state 3, the state in the same topological community as state 4 (Fig 14B; 24 out of 32 participants, $p = 0.007$, two-tailed binomial test), as the model predicts (Fig 14C; 24 out of 32 simulated participants, $p = 0.007$). Topological structure is the only driver of hierarchy discovery in this case, since there is only a single reward and no tasks. The structure of the graph favors clustering state 4 together with state 3 since they belong to the same community. The higher-than-average reward of state 4 then drives up the average reward $\theta$ for that cluster, which in turn drives up average reward $\mu$ for the states that belong to it, and in particular for state 3. In contrast, state 7 often ends up in a separate cluster which is not influenced by the reward of state 3 and thus has an expected reward of 15, the average $\bar{\bar{\theta}}$ for the entire graph.

## Experiment seven: Rewards and planning

While the results of experiment six validated the reward assumptions of our model, they could also be accounted for by alternative models, such as the successor representation [32, 33] or a Gaussian process with a diffusion kernel [26, 31]. We therefore sought to test the unique prediction of our model that, in the absence of community or task structure, hierarchical planning will occur over clusters delineated by boundaries in the reward landscape of the environment.

**Participants.** We recruited 174 participants (68 female) from MTurk. Participants were paid $0.50 plus a bonus equal to the number of points earned on a randomly chosen trial in cents (up to $3.00). This encouraged participants to do their best on every trial. The experiment took 9 minutes on average.

**Design.** We asked participants to navigate a graph ("network of gold mines") with the same structure as in experiments one and four (Fig 15A). Unlike the previous experiments, participants performed a mix of randomly shuffled free choice and forced choice trials. On free choice trials, participants started in a random node and could navigate to any node they chose. Once they reached their target node, they collected the reward ("grams of gold") from that node. On forced choice trials, participants had a specified random target node and they could only collect reward from that node, similarly to the tasks in experiments one through five. Free choice trials encouraged participants to learn the reward distribution, while forced

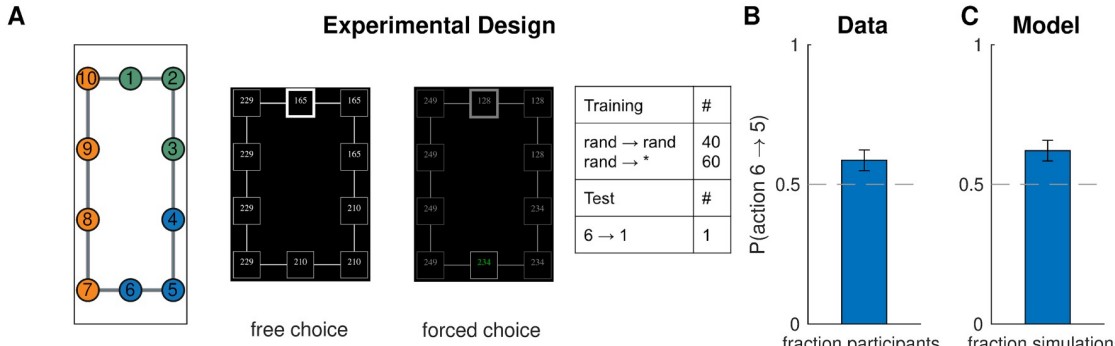

**Fig 15. Rewards induce clusters that influence planning.** A. (Left) Experiment seven employed the same graph as in experiments one and four, with the difference that clusters were induced via the reward rather than the task distribution. (middle) screenshots from free choice and forced choice trials. (Right) training and test protocol. "Rand" indicates that a random state was chosen on each trial, while the asterisk indicates a free choice trial (i.e., the participant was free to choose any node). B. Results from experiment seven showing that participants were more likely to prefer the path with fewer reward cluster boundaries. Error bars are s.e.m. (174 participants). C. Results from simulations showing that the model exhibited the same preference. Notation as in B.

choice trials encouraged planning and prepared participants for the test trial, which was also a forced choice trial. Crucially, the rewards favored a clustering like the one in experiments one and four: states 1,2,3 always delivered the same reward, as did states 4,5,6 and states 7,8,9,10. Similarly to experiments one and four, participants were tested on a forced choice task from node 6 to node 1.

**Procedure.** Participants were told they would navigate a virtual network of gold mines. The graph layout was identical to experiment four (Fig 15A) and participants could see a bird's-eye view of the entire graph, with each node representing a mine and each edge representing a tunnel between mines. Each mine delivered a certain number of points, which were displayed inside the node of the corresponding mine. Similarly to experiment four, participants could navigate between mines using the arrow keys and choose a mine using the space bar, after which they were informed how many points they earned from the chosen mine and immediately began the next trial. Mines always delivered deterministic points as indicated on each mine. Only points earned from the chosen mine counted.

On free choice trials, any mine could be chosen. On forced choice trials, only a single target mine was available and all other minds are grayed out (Fig 15A, right). Attempting to select a different mine resulted in a "incorrect" message flashing and did not change the current mine. There were 60 free and 40 forced choice training trials, starting at random states. The target state on the forced choice trials during training was also random. In order to prevent participants from developing a model-free bias towards one side of the graph based on reward magnitude (for example, because states there happened to be more rewarding), we randomly resampled the points of all states with probability 0.2 on every free choice trial. All reward values were sampled uniformly between 0 and 300.

**Results and discussion.** As in experiments one and four, participants showed a preference for the route with fewer cluster boundaries (Fig 15B; 102 out of 174 participants, $p = 0.03$, two-tailed binomial test). The model made same prediction (Fig 15C; 108 out of 174 participants, $p = 0.001$, two-tailed binomial test). However, this time the clusters were inferred solely based on rewards, rather than topological structure or task distributions. This shows that the reward distribution in the environment can affect hierarchy discovery and consequently planning over that hierarchy, in accordance with our model predictions.

## Discussion

In this study, we proposed a Bayesian model for discovering state hierarchies based on a generative model of the topological structure, rewards, and tasks in the environment. Building on the Bayesian brain hypothesis [35–37] and on principles developed in structure learning [10] and robotics [11], it postulates that the brain learns useful hierarchical representations by inverting this generative model to infer the hidden hierarchical structure of the world. These representations can be subsequently used to plan efficiently and flexibly in the face of changing task demands. The model accounts for a number of phenomena previously reported in the literature and makes new predictions which we verified empirically.

The model goes beyond previous accounts of clustering states in the environment both in the scope of findings it can explain and in the predictions it makes. Schapiro et al. [7] proposed a recurrent neural network that learns community structure based on the temporal statistics of the environment. Their model and other event segmentation models [38] can learn to predict stimuli that tend to co-occur, such as states in the same community, and can detect boundaries when those predictions are violated, for example after a transition to a new community. While this could explain the effects of simulations one through five and possibly experiments one, two and three, it will be challenged by experiments four, five and six, in which the

environment is fully visible and all stimuli co-occur together. More importantly, these models make no predictions regarding action selection, and it is not clear how the learned representations can be used for planning.

One way to implement action selection using the temporal statistics experienced by the agent is to assume that bidirectional associative chains are built based on training sequences and planning is done over low-level states using association-based distances (that is, less closely associated states are considered to require a larger transition cost). This is similar in spirit to the model proposed by Lynn et al. [12] and could replicate the results of experiments one through five. However, the lack of a hierarchical planning component means that it would fail to account for simulations three and four, and the lack of an explicit hierarchy means that simulation two would also pose a challenge. Simulation one controls for the number of transitions, so an associative account would not detect the boundary transitions. It would also fail to capture the reward-based clustering observed in experiment seven.

Solway et al. [8] offer a formal definition of an optimal hierarchy; however, as the authors themselves point out, their analysis is not a plausible account of how hierarchy is learned, because it assumes the agent already knows the optimal solution to all possible tasks in the environment, which defeats the purpose of learning a hierarchy in the first place. A detailed comparison between our model and alternative candidate models that make similar claims is shown in Table 2. Note that there is no need to simulate these models as they categorically cannot capture some of the effects presented here.

The problem of planning has been studied extensively both for biological as well as artificial agents. Correspondingly, our work resonates with several important strands of research, which we discuss in turn below. Refer also to S1 Appendix for further discussion on related ideas.

## Model-based and model-free reinforcement learning

In psychology, planning and goal-directed behavior, which selects actions based on their outcomes, is often contrasted with habitual behavior, which selects actions automatically [39]. The dichotomy between these two kinds of action selection strategies—fast, reflexive, habitual, unconscious responding (also referred to as System 1) on one hand, and slow, reflective, goal-directed, conscious deliberation (also referred to a System 2) on the other—has permeated the field for over a century [40, 41], has been reincarnated in many forms [42–44], and is realized in distinct neural circuits [45–48]. In RL, habitual and goal-directed behaviors have been associated with *model-free* and *model-based* algorithms, respectively [49, 50].

In model-free algorithms, the agent learns a value function for each state and/or each action by trial-and-error, and then chooses actions with respect to the learned values. In model-based RL, the agent learns the reward and transition structure of the environment and combines that information to find actions leading to reward. Model-free RL is associated with fast, reflexive responding because it myopically considers the value of the current state/action only, while model-based RL is associated with slow, reflective responding because it considers the transitions and rewards of multiple states/actions ahead. Despite the superficial similarity of these two systems and the two computational components we propose—a fast online planner, and a slow offline representation learner—there is a profound difference. Both model-free and model-based RL (as well as other versions of habitual and goal-directed strategies) perform online action selection, with agents having to rely on one or the other to solve a particular task. In contrast, our planner makes decisions about which actions to select in a particular task, while the hierarchy discovery process works offline, outside the context of a particular task. In a way, our planner is more like model-based RL in that it performs deliberate, goal-directed

computations based on its internal model of the world (the "model" in model-based RL). The purpose of the hierarchy discovery process is to learn such an internal representation that can be used by a planner with limited computational resources. Indeed, the cognitive limitations inherent in this type of reflective decision-making [43] are the foundational assumptions behind our approach.

Notice that we only invoke a model-based planner in order to motivate hierarchy discovery and to link the hierarchy to decision-making, without committing to the particular HBFS algorithm. While hierarchy discovery can be justified in this way without invoking a model-free system, we certainly do not exclude the possibility of such a system existing and operating in parallel with the planner. A model-free component could easily be incorporated into our framework, for example by having a parallel model-free system which learns the values of states and actions. During decision making, the agent can use a meta-cognitive process to arbitrate between the planner and the model-free system [51]. Like the planner, the model-free system could also operate at different levels of abstraction, learning values for the clusters in addition to the low-level states. This could account for effects of reflexive responding that we explicitly controlled for in our experiments.

Another distinguishing feature of our approach is that, by leveraging the deterministic nature of the transition structure, our planner can rely on simple shortest path algorithms to find solutions to tasks. In contrast, traditional approaches to model-based RL involve computationally costly operations such as value iteration or sampling of candidate trajectories, both of which scale poorly with the size of the state space. Even though the combinatorial explosion of sampling could be managed by heuristics such as pruning of decision trees [29, 52], such approaches are unnecessary in the deterministic domains often considered in planning problems.

It is worth highlighting that our results cannot be explained by standard model-free or model-based RL algorithms. In experiments one, two, four, and five, model-free RL would have learned high values for states 3, 6, and 7, and hence would prefer the transition $6 \rightarrow 7$ on the test trial. In experiments one and four, model-based RL would be indifferent between the two possible trajectories on the test trial due to the symmetry of the graph. In experiments two and five, model-based RL should actually favor the shorter path via $6 \rightarrow 7$. All of these predictions go against our participants' tendency to pick the transition $6 \rightarrow 5$.

The two systems in our proposal might also bear superficial resemblance to the Dyna architecture in RL [53] in which a slow, offline model-based simulator trains a fast, online model-free system to respond adaptively to situations it has never experienced before. This process is reminiscent of hippocampal replay [54] and is particularly useful when the environment changes too often for trial-and-error learning to be effective. In contrast, we propose an offline inference process for learning representations (the "model" in model-based RL) that a model-based system can use to plan in previously unseen tasks. If our proposal is extended with a separate model-free component as suggested above, it could be integrated with Dyna, which would in turn use the model-based system to train the model-free system.

Our results also cannot be explained by the successor representation [33, 55], an intermediate approach along the model-free/model-based continuum that predicts which states a given policy will visit. This will fail to account for experiment three, in which none of the optimal policies visit state 1, and the policies of the random tasks would not favor the transition $6 \rightarrow 7$ over $6 \rightarrow 5$.

## Hierarchical reinforcement learning

A long-standing challenge for traditional RL has been the combinatorial explosion that occurs when planning and learning take place over long time horizons. This challenge been addressed

by hierarchical RL (HRL), which breaks down the problem into sub-problems at multiple levels of abstraction. One influential approach to HRL extends the agent's action repertoire to include *options* [56] which consist of sequences of actions (the option policy) that accomplish certain subgoals (for example, exiting a room). When an option is selected, the corresponding action sequence is executed as a single behavioral unit. Options are also referred to as skills, subroutines, partial policies, macro-actions, or policy chunks, while the original actions are sometimes referred to as primitive actions.

As with regular or "flat" RL, HRL also comes in two distinct flavors [20]: model-free HRL and model-based HRL. Like model-free RL, model-free HRL [57–59] learns a value function; however, in this case the value function is additionally defined for options. Error-driven learning occurs both on the high level by learning which options lead to rewarding outcomes, as well as on the lower level by learning the best option policy for each option. Importantly, there is no notion of a transition function which could be used for planning, and hence model-free HRL could not exhibit the behaviors predicted by our model. Although a model-free component could be incorporated into our framework as discussed above, it would not account for the results of experiments one through five. The only options that could conceivably have been learned by model-free HRL are the ones corresponding to the training tasks (for example, 1 → 3, 4 → 6, and 10 → 7 for experiment one). This set of options could not explain the results of the test trial which requires going in the opposite direction.

In model-based HRL [60], as in model-based RL, the agent separately learns a transition function and a reward function. Additionally, the agent is furnished with an *option model* that specifies the initiation states (for example, locations within a room), termination states (subgoals; for example, doors), average duration, and average reward of each option. The agent can thus plan over options rather than primitive actions, essentially performing mental "jumps" in the state space, allowing it to first form a high-level plan between subgoals reachable via options and then refining that plan on the lower level by simply following the corresponding option policies. This form of *saltatory* model-based HRL is conceptually identical to our proposal. While our work builds on concepts developed in parallel to HRL in the field of robot navigation and planning [11], our model can be cast in HRL terms by considering each task as equivalent to placing a positive reward in the goal state and a small negative reward in all other states, thus encouraging the agent to find the shortest path to the goal state. Edges in the high-level graph $H$ can be seen as options, with subgoals specified by the endpoints of bridges and option policies specifying how to reach the subgoals within a cluster.

Our work introduces two critical improvements to model-based HRL. First, as in model-based RL, model-based HRL assumes planning occurs by sampling trajectories through the state space, which in our proposal is performed by the deterministic and much more efficient HBFS algorithm. Second, as Botvinick et al. [60] point out, a critical open question is how useful options are discovered in the first place. The approach they propose is based on the successor representation [33] under a random walk policy, which partitions the state space along topological bottlenecks. However, this would not predict the results of experiments one through five and experiment seven, in which clustering occurs based on tasks and rewards only.

When framed within HRL, our approach can be viewed as a solution to the option discovery problem which has plagued the field of HRL since its inception, as the original formulation never specified how useful options are learned in the first place. Discovering useful subgoals and, correspondingly, useful options is critical, since an inadequate set of options can lead to dramatically worse performance compared to regular RL [20]. This has led to the proliferation of a rich literature on option discovery, to which we turn next.

## Option discovery

While earlier work on HRL assumed the options are supplied manually [56], a growing number of HRL studies have focused on the problem of discovering useful options. A detailed review of the option discovery literature is beyond the scope of this discussion, but we highlight some of the main approaches. Most option discovery methods fall in one of two broad categories: *state abstraction* and *temporal abstraction* methods [61].

State abstraction methods first decompose the state space by identifying clusters or subgoals, and subsequently identify options based on that decomposition [62, 63]. The state space could be partitioned into clusters based on the value function [64], the state features [65], or the transition function [66, 67]. Other approaches designate certain states as subgoals and learn options that lead to those subgoals, which could be defined by salient events [68], object-object interactions [69], frequently visited states [70–72], or large changes in the reinforcement gradient [72]. Other subgoal discovery methods rely on graph theoretic notions to identify bottlenecks (such as "doors" between rooms) [73], the boundaries of strongly connected regions of the state space (such as the "walls" of rooms) [74], or clusters of states (such as the rooms themselves) [75, 76] as subgoals.

Temporal abstraction (or policy abstraction) methods directly learn the option policies, without resorting to state abstraction as an intermediate step. Some of these approaches identify frequently used action sequences from successful trajectories [61, 77, 78]. Other approaches posit a generative model for policies that favors temporal abstraction, and then perform probabilistic inference to find the optimal policy [79, 80].

Viewed in HRL terms, our model falls into the state abstraction category, since it first partitions the state space into clusters which in turn define subgoals and constrain behavior. However, our model goes beyond these previous attempts: it unifies multiple ideas from these different approaches under a single Bayesian framework that allows it to account for all the behavioral phenomena in our study. To the best of our knowledge, no single option discovery method based on state abstraction would capture all of them. Option discovery methods based on temporal abstraction would also fail to account for the results of experiments one through five, since participants were never trained to navigate in the directions tested in the test trial.

## Future directions

Thus far our model only addresses the question of state chunking (how the environment is represented as discrete states at different levels of abstraction), while leaving open the question of action chunking (how actions are stitched together into larger behavioral units at different levels of temporal abstraction). Fitting the model into the HRL framework described above might seem like one way to incorporate action chunking, by assuming that the options leading to bridge endpoints are like action chunks that, when invoked, delegate behavior to a low-level controller that executes the sequence of actions in the option policy as a single behavioral unit. Yet the metaphor of options as action chunks is not necessarily appropriate; option policies execute in a closed-loop fashion, taking into consideration each state they encounter before choosing the next action. In contrast, action chunks are often operationally defined as open-loop action sequences that disregard intermediate states [57].

An alternative way to accommodate action chunking that more closely adheres to this definition is to allow caching of solutions to repeated calls to the planner with the same arguments [29]. That way, when a certain subpath within a cluster is traversed frequently as part of many tasks (for example, getting from your bed your bedroom door), the corresponding action sequences could be cached and, when the subtask needs to be solved as part of a larger task (for example, getting from your bed to work), the action sequence can be retrieved from the

cache and executed as a single unit, thus removing the need to unnecessarily recompute it every time. Indeed, such a simple scheme could account for phenomena such as action slips [57], or even be used to model task-bracketing activity in striatum (see S1 Appendix). Implementing action chunking in this way could also account for the speeding up of responses during training in our experiments.

Another limitation of our approach is the hard restriction that each low-level node is a member of a single cluster. This restriction could be relaxed by explicitly building in "soft" cluster membership into the generative model, or by allowing agents to entertain multiple possible hierarchies when planning. While we do not exclude the former approach as a possibility, the latter approach arises naturally from our generative model as the posterior can represent graded beliefs in multiple hypotheses. This is consistent with previous work showing how mixtures of deterministic representations can account for graded effects [81].

Our treatment of hierarchy discovery is restricted to the computational level of analysis (in the Marrian sense) [27] and assumes the agent can accurately sample from the posterior over hierarchies $P(H|D)$. Scaling up the model beyond the toy experiments considered here would require relaxing this assumption and instead resorting to approximate Bayesian inference. This points to a fruitful avenue for future research, namely probing the algorithmic details of hierarchy discovery. One possibility is to take our Monte Carlo sampler as a process model of hierarchy inference in the brain. This is consistent with previous work casting human probabilistic inference as a form of Monte Carlo sampling [82–87]. Of particular relevance are accounts of theory acquisition during development as a form of stochastic search in theory space [88]. In this light, hierarchy discovery can be understood as stochastic search in the space of hierarchies that occurs throughout development and aims to distill the complex structure of the world into a compact representation that can be used for nimble decision-making. This algorithmic account predicts a bias towards the prior and autocorrelation in the sampled hierarchies that could be assessed empirically. A noisy approximate inference process could also potentially explain the quantitative discrepancy between our model and the participants in experiment three.

Even with a sampling approximation, inference in large-scale environments would pose a challenge to computing the (unnormalized) posterior, which requires iterating over all nodes in the low-level graph. This means that the complexity of a single update of a single cluster assignment is $O(N)$, which is very inefficient even for an offline process. This computational burden can be alleviated by noticing that updating the cluster assignment of a given node mainly affects the posterior through the immediately adjacent nodes. Thus instead of computing the posterior from scratch for each node update, the agent can simply adjust the posterior based on local changes (in fact, a Metropolis-Hastings rule like the one we use here can allow the agent to completely ignore the rest of the posterior and only consider local changes). Such a factorization of the posterior would bring the computational complexity of a single node update to $O(1)$. Furthermore, the Gibbs sampler we use here allows updates to be performed in any order, without sacrificing convergence guarantees. Overall, this suggests an ecologically appealing account of hierarchy discovery, according to which agents organize their immediate surroundings into clusters using only local update rules, without considering the rest of the world, yet in doing so they form global representations that can facilitate efficient planning between states that they never perceive in a single instant. The only aspect of the posterior which cannot be factorized is the connectivity constraint (see Methods), which could be addressed, for example, by assigning nodes to adjacent clusters only. Such a cheap and efficient online approximation might also explain how people were able to perform hierarchy inference during our relatively short experiments.

There are several ways in which our model could be extended to support planning in richer, large-scale environments. One way would be to allow deeper hierarchies ($L > 2$), which would be necessary in order to maintain the computational efficiency of the planner as the size of the graph increases. This could be achieved by recursively clustering the high-level states in $H$ into higher-level states in another graph $H'$, then clustering those in yet another graph $H''$, and so on up to hierarchy depth $L$ that could be prespecified or also inferred from the data. The hierarchical planner can similarly extend recursively by reusing the same logic at the higher levels.

Yet even with deep hierarchies, the learned representations would pose a challenge to the planner as a number of low-level states in $G$ grows to the scale encountered in real life (indeed, it would also pose a challenge to the long-term storage system). This highlights another limitation of our model, namely the use of tabular representations in which each state is represented by a separate token. This could be overcome by recognizing that there is redundancy in the environment [89]—that is, the local structure of the graph $G$ can be highly repetitive, with the same theme occurring over and over in different parts of the graph. For example, most cities have streets and buildings, most buildings have rooms and hallways, and most rooms have similar layouts. Representing each and every part of the environment would thus be wasteful, and this redundancy can be exploited by introducing templates or modules—blueprints for clusters that can compress the hierarchical representation by extracting the shared structure across clusters of the same type and only representing differences from some prototypical cluster. Clustering these modules will naturally give rise to the kind of compositionality characteristic of natural environments. Combined with partial observability and deeper hierarchies, this would allow the model to learn representations that support efficient planning in environments that approach the real world in their scale and complexity. Finally, the requirements that $G$ is unweighted and undirected can be lifted if the planner is a hierarchical extension of a more sophisticated shortest path algorithm, such as Dijkstra's algorithm [17].

## Conclusion

In summary, we propose a normatively motivated Bayesian model of hierarchy discovery for planning. The model builds on first principles from the fields of structure learning and robotics, and recapitulates a number of behavioral effects such as detection of boundary transitions, identification of bottleneck states, and surprise to transitions across clusters. The novel predictions of the model were validated in a series of behavioral experiments demonstrating the importance of the task and reward distributions in the environment, which could bias the discovered hierarchy in a way that is either beneficial or detrimental on new tasks. We also showed that the model accounts for reward generalization and uncertainty-based learning effects in a way that is consistent with human behavior. Together, these results provide strong support for a computational architecture in which an incremental offline process infers the hidden hierarchical structure of the environment, which is then used by an efficient online planner to flexibly solve novel tasks. We believe our approach is an important step towards understanding how the brain constructs an internal representation of the world for adaptive decision making.

## Methods

### Ethics statement

All experiments involved human participants and were approved by the Harvard Institutional Review Board, number IRB15-2048. Participants that performed experiments on Amazon MTurk gave written consent (experiments one through five and experiment seven). Consent was not obtained for experiments six and eight since the data were analyzed anonymously.

## Inference

We frame hierarchy discovery as computation of the posterior probability distribution $P(H|D)$. Since computing the posterior exactly is intractable, we approximate Bayesian inference over $H$ using Metropolis-within-Gibbs sampling [90], a form of Markov chain Monte Carlo (MCMC). We initialized $H$ by sampling from the prior $P(H)$ and on each MCMC iteration, we updated each component of $H$ in turn by sampling from its (unnormalized) posterior conditioned on all other components in a single Metropolis-Hastings step, in the following order:

1. update the cluster assignments $c$ using the conditional CRP prior (algorithm 5 in [91]),

2. update $p$, $q$, $p'$, $p''$ using a truncated Gaussian random walk with standard deviation 0.1 (i.e. the proposal distribution is a Gaussian centered on the old value, excluding values above 1 or below 0),

3. update the hierarchical edges $E'$ with a proposal distribution that randomly flips the presence or absence of each edge with probability 0.1,

4. update the average cluster rewards $\theta$ using a Gaussian random walk with standard deviation 1, and

5. update the average state rewards $\mu$ using a Gaussian random walk with standard deviation 1.

The samples generated in this way approximate draws from the posterior, with asymptotic convergence to the true posterior in the limit of infinite samples. This approach can also be interpreted as stochastic hill climbing with respect to a utility function defined by the posterior, which has been previously used to find useful hierarchies for robot navigation [11]. Note that while we chose to sample the components of $H$ in an order that roughly follows the generative model, the exact order is actually irrelevant because we sample to convergence and only take the final sample, and hence any arbitrary order would yield similar results.

In all five simulations and experiments one through seven, for each simulated participant, we generated a Markov chain of a given length (see Table 1) and used the final sample $H$ to generate predictions. This can be viewed as a form of probability matching over hierarchies, and is consistent with psychologically plausible algorithms for hypothesis generation and updating, although elucidating the algorithmic details of hierarchy discovery is beyond the scope of our present work. For experiment three, we ran the MCMC sampler before each probe trial, while for the other experiments we ran it before the test trial. We did this for purposes of efficiency, since we only evaluated predictions on the probe/test trials, and since our computational-level analysis is agnostic to the algorithmic details of hierarchy discovery.

In order to estimate entropy in experiment eight, we used a separate MCMC sampler for each graph for each participant to generate a set of $M = 50$ samples, using a lag of 100 iterations and burn-in period of 5000 iterations (for a total of 10000 generated samples, as in all other simulations).

Additionally, HBFS requires that the subgraph induced in $G$ by each cluster must form a single connected component; that is, for every pair of states $(u, v)$ in a cluster $w = c_u = c_v$, there must exist a path $(u, x_1, \ldots, x_k, v)$ such that $c_{x_k} = w \; \forall k$. To enforce this constraint, we imposed a penalty by subtracting 100 from the (unnormalized) log posterior for each pair of states in the same cluster that are not connected by a path passing through the cluster. This is equivalent to augmenting the generative model with the following rejection sampling procedure: draw $H$ according to $P(H|D)$ and for each such pair of disconnected states, perform a Bernoulli coin flip with probability of success equal to $e^{-100}$. If all coin flips are successful, keep $H$, otherwise

repeat the process with a new $H$. By imposing a "soft" constraint in this way, we ensured that the sampling algorithm can recover from a bad initialization by incrementally adjusting pairs of nodes that violate the constraint. Note that this still results in a valid posterior since normalization is not necessary for approximate inference.

## Decision making

We assume choices based on $H$ are either optimal (according to the model) with probably $\epsilon$, or random with probability $1 - \epsilon$. This is similar in spirit to the $\epsilon$-greedy algorithm in RL, with the difference that we assume the meta-choice to choose randomly occurs before computing the optimal answer. This is equivalent to assuming that on $1 - \epsilon$ of trials, participants simply do not perform the necessary computation. Such lapses could be due to a number of reasons, such as inattention or fatigue. While other factors such as motor variability may also contribute to choice stochasticity, we allow those to be absorbed by the $\epsilon$ parameter and leave them as the potential subject of future work.

Another source of variance in choices is the sampled hierarchy $H$, which could be different for each simulated participant. While our current experiments prohibit direct comparisons between hierarchies inferred by participants and hierarchies inferred by the model, any systematic variation in the sample hierarchies (for example, due to uncertainty of the posterior, as in experiments three and eight) would manifest when choices are aggregated across participants, as in our analyses.

**Hierarchical planning.** The optimal algorithm to find the shortest path between a pair of low-level vertices $(s, g)$ in $G$ is breadth-first search (BFS) [17] whose time and memory complexity is $O(N)$ (assuming $O(1)$ vertex degrees, i.e. $|E| = O(N)$). We use a natural extension of BFS to hierarchical graphs (hierarchical BFS or HBFS) [11] that leverages $H$ to find paths more efficiently than BFS (approximately $O(\sqrt{N})$ time and memory). Intuitively, HBFS works by first finding a high-level path between the clusters of $s$ and $g$, $c_s$ and $c_g$, and then finding a low-level path within the cluster of $s$ between $s$ and the first bridge on the high-level path.

In particular, HBFS first finds a high-level path $(w_1, \ldots, w_{m'})$ between $c_s$ and $c_g$ in the high-level graph $H$ (note that $w_1 = c_s$ and $w_{m'} = c_g$). Then it finds a low-level path $(y_1, \ldots, y_m)$ between $s$ and $u$ in $G[S]$ (note that $s = y_1$ and $u = y_m$), where $(u, v) = b_{w_1, w_2}$ is the first bridge on the high-level path, $S = \{x : c_x = c_s\}$ is the set of all low-level vertices in the same cluster as $s$, and $G[S]$ is the subgraph induced in $G$ by $S$. HBFS then returns $y_2$, the next vertex to move to from $s$, or, alternatively, full the path to the next cluster, $(y_2, \ldots, y_m)$.

In an efficient hierarchy, the number of clusters will be $|V'| = O(\sqrt{N})$ and the size of each cluster $w$ will also be $n_w = O(\sqrt{N})$, resulting in $O(\sqrt{N})$ time and memory complexity for HBFS. Note that actually traversing the full low-level path from $s$ to $g$ in $G$ still takes $O(N)$ time; HBFS simply computes the next step, ensuring the agent can progress towards the goal without computing the full low-level path in advance (in our simulations, we actually computed the full path in order to simulate execution in addition to planning). HBFS can straightforwardly extend to deeper hierarchies with $L > 2$, with the corresponding complexity becoming $O(\sqrt[L]{N} + L)$.

**Algorithm 1** HBFS($s$, $g$, $H$, $G$)

```
1: path′← BFS(cₛ, c_g, (V′, E′))
2: path ← []
3: for all (w, z) in path′ do
4:    (u, v)←b_{w,z}
5:    S ← {x: cₓ = cₛ}
6:    append(path, BFS(s, u, G[S]))
7:    append(path, (u, v))
```

```
 8:     s ← v
 9: end for
10: S ← {x: c_x = c_s}
11: append(path, BFS(s, g, G[S]))
12: return path
```

The pseudocode for HBFS used in our simulations is shown in Algorithm 1. Our particular implementation of HBFS takes as arguments the starting state $s \in V$, the goal state $g \in V$, the hierarchy $H$ and the low-level graph $G$. The variables $c$ and $b$ refer to the cluster assignments and bridges in $H$, respectively. Note that $s$ changes after each iteration. We assume the existence of a function BFS which takes as arguments a starting state, a goal state and a graph and returns the shortest path between those states as a list of edges.

Note that we are not making any specific commitments to the cognitive plausibility of HBFS, and that any other hierarchical planner based on shortest paths would make similar predictions. Also note that HBFS could be straightforwardly extended to deeper hierarchies by introducing a depth level parameter $l$ and recursively calling HBFS instead of BFS on line 1 if $l > 2$. Finally, note that HBFS as implemented here still requires $O(N)$ time and memory as it finds the full path in $G$. For returning only the first few actions, the for-loop could be interrupted after the first iteration, which would yield the $O(\sqrt[l]{N}+L)$ complexity.

**Rewards.** To model reward generalization in experiment six, we set $\bar{\theta} = 15$ and $r_{4,1} = 30$, in accordance with the experimental instructions. The node selected by the simulation was the one with the greater (approximate) expected reward $\mathbb{E}[r_s|D] \approx \theta_{c_s}$, where $\theta$ are the cluster rewards and $c$ are the cluster assignments in the sampled hierarchy $H$.

To model cluster inferences based on rewards in experiment seven, we only simulated a single training trial and set $r_{1,1} = r_{2,1} = r_{3,1}$, $r_{4,1} = r_{5,1} = r_{6,1}$, and $r_{7,1} = r_{8,1} = r_{9,1} = r_{10,1}$, with the specific values chosen at random between 0 and 30 (we scaled down the rewards experienced by participants by a factor of 10). Since the hierarchy discovery algorithm has no notion of maximizing reward (it merely treats rewards as features), the magnitude of the rewards is irrelevant. We did not model the random changes of reward values throughout training, which were introduced purely for control purposes and are irrelevant for hierarchy discovery as framed here. Note that the model could easily accommodate dynamic rewards by assuming drifting $\mu$'s, however this would unnecessarily complicate the model without making substantial contributions to the core theoretical predictions. As in experiment six, we set $\bar{\theta} = 15$, the expected reward based on the instructions.

## Supporting information

**S1 Appendix. Supplemental results, methods, and discussion.**
(PDF)

## Acknowledgments

We are grateful to Finale Doshi-Velez, George Konidaris, Bence Ölveczky, Jan Drugowitsch, and Nao Uchida for the helpful discussions.

## Author Contributions

**Conceptualization:** Momchil S. Tomov, Samuel J. Gershman.

**Data curation:** Momchil S. Tomov, Samyukta Yagati, Agni Kumar, Wanqian Yang.

**Formal analysis:** Samyukta Yagati, Agni Kumar.

**Funding acquisition:** Samuel J. Gershman.

**Investigation:** Momchil S. Tomov.

**Methodology:** Momchil S. Tomov.

**Project administration:** Momchil S. Tomov.

**Software:** Momchil S. Tomov, Samyukta Yagati, Agni Kumar, Wanqian Yang.

**Supervision:** Samuel J. Gershman.

**Validation:** Momchil S. Tomov.

**Visualization:** Momchil S. Tomov, Samyukta Yagati, Wanqian Yang.

**Writing – original draft:** Momchil S. Tomov.

**Writing – review & editing:** Momchil S. Tomov, Samyukta Yagati, Agni Kumar, Wanqian Yang, Samuel J. Gershman.

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
