## [Decision Letter · Decision Letter 0]

9 Oct 2019

Dear Dr. Tomov,

First of all let me apologise for the exceptional delay, due to the unfortunate overlap of virtually all of the single reasons that could delay the process. Thank you very much for submitting your manuscript 'Discovery of Hierarchical Representations for Efficient Planning' for review by PLOS Computational Biology. Your manuscript has been fully evaluated by the PLOS Computational Biology editorial team and in this case also by independent peer reviewers. The reviewers appreciated the attention to an important problem, but raised some concerns about the manuscript as it currently stands. These involve mainly a plausible competing model that should be ruled out in addition to the flat association-based model, the ambiguity of one experiment and the related interpretations ( Reviewer #1 ) and the length and organization of the manuscript ( Reviewer #2 ), along with minor details. While your manuscript cannot be accepted in its present form, we are willing to consider a revised version in which the issues raised by the reviewers have been adequately addressed. We cannot, of course, promise publication at that time.

Sincerely,

David Pascucci, Ph.D.

Guest Editor

PLOS Computational Biology

Daniele Marinazzo

Deputy Editor

PLOS Computational Biology

[LINK]

Reviewer's Responses to Questions

**Comments to the Authors:**

Reviewer #1: Planning over a large state space is computationally demanding. The computational cost of planning can be reduced with hierarchical representations, and in certain cases, when the right representations are chosen, these computational savings can come without performance substantial costs to performance. However, the question of how people should and do identify the right abstract representations is an open and important problem. Here Tomov and colleagues address this question by developing a generative hierarchical model of low level states and their transition structure, and inverting this generative graph with to do inference over the abstract representations giving rise to the observed low-level state transitions. The model, is able to capture an impressive array of foundational behavioral findings from previous experiments. In addition, the authors test an impressive array of new model predictions with new experiments in human participants, and, by and large, find that the model provides a good qualitative match to the data.

This work is timely, interesting, and undoubtedly very thorough. Given the amount of material, it is probably no surprise that I have a few concerns with the current manuscript, but my impression is that with some changes this paper will make an impressive contribution to an active and important area of computational neuroscience. A complete listing of my concerns is below.

In several places the authors rule out a flat association-based model – however I have the sense that the authors were really rejecting a straw-man model – one where the associations are simply between two consecutive actions or states. As I read the manuscript I had another, slightly more complex associative model in mind that I don’t think is ruled out by the new experiments. In particular, bidirectional associative chains are built based on training sequences and planning is done over low-level states but using association-based distances (eg. less closely associated states are considered to require a larger transition cost). I understand that this idea wouldn’t solve the computational problems associated with planning, but I’m curious whether they can really rule something like this out in the new behavioral experiments that they provide.

How were the model clustering updates made in the learning task? Was the MCMC re-run completely with each new data point? Clearly, in the human experiment, people are adjusting their abstract state clustering online. The authors seem to pitch the model as something that would occur offline, due to heavy computational demand, and I guess in some sense I wondered whether the ability of people to adjust their posterior over state clustering might actually be evidence against this computationally intensive hierarchical discovery procedure, and instead suggest that something cheaper and simpler is done online.

I don’t see the active learning experiment as providing evidence for the proposed model. I’ve got a few concerns with the interpretation of this experiment. The first is with the premise; based on the description of the experiment I don’t understand why participants would report the subway line that has the largest effect on high level clustering. It seems more reasonable that people would choose the line that has the greatest potential to reduce the average shortest path between states -- and this appears relatively consistent with what participants did. I’m also not convinced that the predictions of the model line up all that well with the data, especially considering the large number of alternative metrics that could be used to provide results similar to the subject data. Given the quality of the rest of the experiments, and the length of the paper, I would actually suggest that the authors remove this section from the manuscript… and if they don’t they should provide a more honest interpretation of the experimental results, which in my mind don’t provide clear evidence for the proposed model.

I’m a bit confused about how the cross-cluster edges in G are conditionally independent of b (the bridges) – I would think that the existence of a bridge necessitates a cross-cluster edge, would it not?

In general, I found the generative model to be a plausible abstraction of real world environments. However, one aspect that I feel uncomfortable with is the hard restriction on group membership (each low level state is part of one and only one high level state). I’m curious whether the authors considered a more continuous (or even compositional) grouping strategy and if so, whether the hard restriction makes different predictions than these alternatives.

I’m unclear on what the lower case c’s are in sampling statement 13 – it would be helpful if they could be defined and that sampling statement unpacked in the text.

The authors set the chinese restaurant prior to be 1, favoring a relatively small number of high level clusters. I’m curious why the authors wouldn’t just infer the CRP prior from the data, rather than set it as a parameter. If it were inferred, rather than fixed, would it take a larger value?

The behavioral result modeled in simulation 5 was unclear from the description. Participants hit the same button no matter what image they see? I’m not sure I understand why RT under such conditions should relate to the number of abstract state boundaries crossed. There were also some typos in figure legend (8b) so maybe this just requires some clarification.

Page 68: “in all stimuli co-occur together” → but don’t they sequentially fixate them?

Page 78: I don’t understand why a successor representation would not predict results of the experiments. If a successor stores the state-state transitions – shouldn’t it pick up on the over-representation of within cluster transitions during training? There are some directional issues (eg. when training/testing require going in opposite directions) – so maybe that is why the authors think they couldn’t account for them?

It would be useful if the authors could provide more description of the variables and generative structures in figure 3 B,C&D in the figure captions (rather than just in the text).

The term “density” is introduced on page 17 but only explained after the sampling statements are provided several pages later.

Reviewer #2: The authors propose to cast human hierarchical planning as an on-line visiting - essentially, a multi-level breadth-first procedure - on a non-parametric probabilistic model of the environment and its latent causes structured in clusters on the base of topological properties, reward and task distribution.

The learning of model's structure precedes - maybe, it would be more accurate to say that it proceeds in the background like a continuous updating - the planning execution; the authors assume that a Gibbs-like sampling inferential process can exist that intuitively requires more massive computational resources and longer execution time.

Consistency and plausibility of the approach have been pointed out using both a set of simulations taken from the neuroscientific literature and a series of behavioural experiments where collected data have been statistically compared to the computational results.

This reviewer has just one primary concern, one kind suggestion and a couple of other scientific curiosities (maybe doubts).

Concern: the paper is well-written but, definitely, too long. One would try to group the most significant points of the discursive sections in "General Discussion" and to leave introductive and more descriptive parts to the Supplemental Material, although they are appreciable. Presumably, the paper will likely gain in readability and will remain focused on its principal scientific purpose.

Suggestion: The idea to rethink abstraction as an inferential process makes sense, and it is valuable the attention paid in making sure that the theoretical proposal still has a physiological plausibility. Nevertheless, it is a non-binding opinion of this reviewer that the choice of parameters'values, as well as their potential analogy with biological/physiological correlates, deservers to be furtherly illustrated. In this sense, some details could integrate the explanation of the adopted framework.

Curiosities/doubts:

1) One criterion conditioning on cluster determination is the topological structure of the environment. The authors posit that observations topologically highly connected need to be in the same cluster. Although this principle is as useful as intuitive, there might be other properties more salient to take into account, mainly if the goal of the hierarchical model is planning. For instance, a node bridging two different communities belongs to any path connecting them, independently to the couple di nodes chosen as start and goal. In this view, the topological saliency of that node is remarkable.

How much is the presented framework configurable in that sense?

2) Authors have fixed an order to update the posterior conditioned on all other H's component. Does not Gibbs sampling assume exchangeable the posterior to approximate? What does that order entail?

3) Are Schapiro and Solway (one and two through five) both indispensable? Do not they show similar points?

**Have all data underlying the figures and results presented in the manuscript been provided?**

Reviewer #1: Yes

Reviewer #2: Yes

PLOS authors have the option to publish the peer review history of their article (what does this mean?). If published, this will include your full peer review and any attached files.

Reviewer #1: No

Reviewer #2: No

---

## [Decision Letter · Decision Letter 1]

20 Nov 2019

Dear Dr Tomov,

Thank you very much for submitting your revised manuscript, 'Discovery of Hierarchical Representations for Efficient Planning', to PLOS Computational Biology. Based on the advice provided by one of the two reviewers, the manuscript requires still some minor changes before being accepted for publication. These include the justification of the order chosen for the Gibbs sampling procedure and the revision and correction of some remaining typos.

We would therefore like to ask you to modify the manuscript according to the review recommendations before we can consider your manuscript for acceptance. Your revisions should address the specific points made by the reviewer and we encourage you to respond to particular issues. Please note while forming your response, if your article is accepted, you may have the opportunity to make the peer review history publicly available. The record will include editor decision letters (with reviews) and your responses to reviewer comments. If eligible, we will contact you to opt in or out.raised.

- Supporting Information uploaded as separate files, titled 'Dataset', 'Figure', 'Table', 'Text', 'Protocol', 'Audio', or 'Video'.

We hope to receive your revised manuscript within the next 30 days. If you anticipate any delay in its return, we ask that you let us know the expected resubmission date by email at ploscompbiol@plos.org.

Sincerely,

David Pascucci, Ph.D.

Guest Editor

PLOS Computational Biology

Daniele Marinazzo

Deputy Editor

PLOS Computational Biology

[LINK]

Reviewer's Responses to Questions

**Comments to the Authors:**

Reviewer #1: The authors have addressed all of my concers.

Reviewer #2: Substantially, the authors have satisfactorily accepted the main proposed suggestion. Specifically, they decided to move the eighth experiment and a section of discussion - the choice of moving the action chunking section seems very appropriate - to the manuscript's supplement.

However, this reviewer would need a clarification concerning the order chosen for the Gibbs sampling. Established that it can be performed in any order, even though the specified order influences the stochastic process, e.g., determining different mixing times and so affecting efficiency, the question is: why did the authors use this particular order? Would have been equal to start the updating procedure by flipping the presence/absence of an edge in E′? Is there any inferential reason?

This reviewer thinks it is worth to spend a brief comment about this issue, for the sake of completeness.

A few typos persist in the text; for instance, in Tab. 2 remains Experiment 8 and presumably should be mentioned its displacement in the supporting information text; in the caption of Fig. 8b it is probably "more slowly" in the place of "more slower"; page 79, item 3, "update " is repeated.

**Have all data underlying the figures and results presented in the manuscript been provided?**

Reviewer #1: None

Reviewer #2: Yes

PLOS authors have the option to publish the peer review history of their article (what does this mean?). If published, this will include your full peer review and any attached files.

Reviewer #1: No

Reviewer #2: No

---

## [Editor Report · Decision Letter 2]

10 Dec 2019

Dear Dr Tomov,

We are pleased to inform you that your manuscript 'Discovery of Hierarchical Representations for Efficient Planning' has been provisionally accepted for publication in PLOS Computational Biology.

In the meantime, please log into Editorial Manager at https://www.editorialmanager.com/pcompbiol/, click the "Update My Information" link at the top of the page, and update your user information to ensure an efficient production and billing process.

One of the goals of PLOS is to make science accessible to educators and the public. PLOS staff issue occasional press releases and make early versions of PLOS Computational Biology articles available to science writers and journalists. PLOS staff also collaborate with Communication and Public Information Offices and would be happy to work with the relevant people at your institution or funding agency. If your institution or funding agency is interested in promoting your findings, please ask them to coordinate their releases with PLOS (contact ploscompbiol@plos.org).

Thank you again for supporting Open Access publishing. We look forward to publishing your paper in PLOS Computational Biology.

Sincerely,

David Pascucci, Ph.D.

Guest Editor

PLOS Computational Biology

Daniele Marinazzo

Deputy Editor

PLOS Computational Biology